

# Dissipation ratio and eddy diffusivity of turbulent and salt finger mixing derived from microstructure measurements

Jianing Li[1], Qingxuan Yang[1, 2, 3], Hui Sun[1]

[1]College of Oceanic and Atmospheric Sciences, Ocean University of China, Qingdao, China
[2]Physical Oceanography Laboratory/Institute for Advanced Ocean Study/Sanya Oceanographic Institution, Ocean University of China, Qingdao/Sanya, China
[3]Laboratory for Ocean Dynamics and Climate, Pilot National Laboratory for Marine Science and Technology, Qingdao, China

*Correspondence to*: Q. Yang (yangqx@ouc.edu.cn)

**Abstract.** Eddy diffusivity is usually estimated by using the Osborn relation assuming a constant dissipation ratio of 0.2. In this study, we examine dissipation ratios and eddy diffusivities of turbulent mixing and salt finger mixing based on microstructure datasets. We find the dissipation ratio of turbulence $\Gamma^T$ is highly variable with a median value clearly greater than 0.2, which shows strong seasonal variation and decreases slightly with depth in the western equatorial Pacific, but obviously increases in vertical in the midlatitude Atlantic. $\Gamma^T$ is jointly modulated by the Ozmidov scale to the Thorpe scale

ratio $R_{OT}$ and the buoyancy Reynolds number $Re_b$, namely $\Gamma^T \propto R_{OT}^{-4/3} \cdot Re_b^{1/2}$. The eddy diffusivity based on observed $\Gamma^T$ is larger than that estimated with 0.2, and presents a much stronger bottom enhancement. The eddy diffusivities of heat and salt for salt finger are calculated by two "analogical" Osborn equations; and their corresponding "effective" dissipation ratios $\Gamma_\theta^F$ and $\Gamma_S^F$ are explored. $\Gamma_\theta^F$ scatters over two orders of magnitude with a median value of 0.47, and is mostly linearly correlated with $\Gamma_S^F$ as $\Gamma_S^F \approx 5\Gamma_\theta^F$. The density flux ratio for salt finger decreases sharply with density ratio $R_\rho$ smaller than 2.4 but

regrows to a larger value with $R_\rho$ exceeding 2.4. The salt finger-induced eddy diffusivities become more comparable or even stronger than the turbulent diffusivities with depth. This study highlights the influences of variable dissipation ratios and different mixing types on eddy diffusivity estimates, and should help further improvement of mixing estimate and parameterization.

## 1 Introduction

Microscale turbulence in the ocean is patchy and intermittent. Compared with molecular diffusion, it mixes materials in a larger scale with a higher efficiency, playing a leading role in re-distributing heat (Pujiana et al., 2018), dissolved gases (Sabine et al., 2004), pollutants (Kukulka et al., 2016), nutrients and plankton (Whitt et al., 2017), thus shaping ocean general circulations and influencing bio-chemical processes in the ocean (Wunsch and Ferrari, 2004). These effects impact global environment and climate change (Jackson et al., 2008).



Due to these significant effects of microscale mixing, the outputs of ocean general circulation and climate models are deeply affected by the mixing intensity and variation (Jayne, 2009). Since the grid size is too coarse to resolve microscale processes, mixing parameterizations are mostly used as a proxy of turbulence effects in such models (Klymak and Legg, 2010). The proposing, verification and development of mixing parameterizations heavily rely on our perceptions of mixing intensity and spatiotemporal variation observed in the real ocean. Therefore, to accurately estimate eddy diffusivity based on observations

has always been an unremitting pursuit of researchers. On one hand, many parameterization methods are developed and widely used to infer eddy diffusivity (e.g., GHP scaling, Gregg et al., 2003; MG scaling, MacKinnon and Gregg, 2003; and the Thorpe scale method, Dillon, 1982), thanks to abundant accumulation of traditional hydrographic observations. These methods yield a mediocre estimate based on fine-scale profiles of temperature and/or velocity with resolution significantly larger than microscale, and may have applicability problems induced by different mechanisms and hydrologic conditions

(Mater et al., 2015). On the other hand, microstructure measurements provide a much more accurate estimate of turbulence behaviors (St. Laurent et al., 2012), although the amount of data is relatively small. With the development of observation technology and the advancement of instruments, microstructure data is experiencing a rapid growth. However, neither parameterizations nor microstructure measurements can directly provide eddy diffusivity values; what they infer is the dissipation rate of turbulent kinetic energy (TKE) $\varepsilon$. Assuming mixing is driven by turbulence, the eddy diffusivity of density

is then estimated by the conventional Osborn relation, $K_\rho = \frac{R_f}{1-R_f} \cdot \frac{\varepsilon}{N^2}$ with $R_f/(1-R_f)=\Gamma^T=0.2$ (e.g., St. Laurent et al., 2012), where $R_f$ is flux Richardson Number, $N^2$ is buoyancy frequency squared, and $\Gamma^T$ is the dissipation ratio of turbulence.

However, there are two inadequacies in the application of the Osborn relation. First, the value of $\Gamma^T$ should be carefully inspected. In the frame of steady, homogeneous turbulence, a balance between TKE production ($P$), buoyancy flux ($B$) and dissipation can be reached, $P+B-\varepsilon=0$. And $\Gamma^T$ is the ratio of the buoyancy flux to the dissipation, $B/\varepsilon$, which describes the

relative proportion of how much TKE is converted to potential energy and irreversibly dissipated to heat. Combining limited measurements with theoretical prediction, Osborn (1980) took the critical value of $R_f$ as $R_f \leq 0.15$, resulting in $K_\rho < 0.2\varepsilon/N^2$. Following that, $\Gamma^T$ is usually taken as a constant of 0.2. Eddy diffusivities of heat ($K_\theta$), salt ($K_S$) and density are equal for turbulent mixing, so these diffusivity values can be easily determined by the Osborn relation as long as $\Gamma^T$ is accurately measured. $\Gamma^T \approx 0.2$ is confirmed to be reasonable by some observations (Gregg et al., 2018); however, besides findings from

laboratory experiments and direct numerical simulations (Barry et al., 2001; Jackson and Rehmann, 2003; Shih et al., 2005; Salehipour et al., 2016), there are considerable and accumulating observational evidence indicating $\Gamma^T$ is significantly variable in both space and time, with a variation range covering several orders of magnitude, typically from $10^{-2}$ to $10^{1}$ (Moum, 1996; Smyth et al., 2001; Mashayek et al., 2017; Ijichi and Hibiya, 2018; Monismith et al., 2018; Vladoiu et al., 2021; Li et al., 2023).

Observations conducted in different regions showed the statistical feature of $\Gamma^T$ is significantly distinct from region to region, and the repeated measurements at some locations suggested $\Gamma^T$ is obviously greater than 0.2 (Ijichi and Hibiya, 2018), indicating taking $\Gamma^T=0.2$ could significantly underestimate eddy diffusivity in these regions. Besides, microstructure



measurements from both upper layer and the whole water column suggested $\Gamma^T$ generally increases with depth, by as much as an order of magnitude (Ijichi and Hibiya, 2018; Li et al., 2023). Thus, taking $\Gamma^T$ as a constant also leads to an

underestimate of eddy diffusivity in the deep layer. These underestimated eddy diffusivities may be a part of the answer to "the missing mixing" puzzle (Wunsch and Ferrari, 2004). Some studies do show that the magnitude and pattern of $\Gamma^T$ plays a key role in regulating global ocean general circulation (Mashayek et al., 2017; Cimoli et al., 2019). Moreover, $\Gamma^T$ is reported to be modulated by turbulence features and is closely correlated with several parameters describing turbulence state, such as turbulence "age" $R_{OT}$ (the ratio of the Ozmidov scale to the Thorpe scale; Ijichi and Hibiya, 2018) and turbulence "intensity"

$\mathrm{Re}_b$ (buoyancy Reynolds number; Mashayek et al., 2017). However, different correlations between $\Gamma^T$ and these parameters are found in different regions. Taking $\mathrm{Re}_b$ as an example, different studies concluded that their relation could be negatively correlated (Monismith et al., 2018), nonmonotonically correlated (Mashayek et al., 2017), or uncorrelated (Ijichi and Hibiya, 2018). In a word, taking $\Gamma^T$ as a constant of 0.2 brings a large bias into eddy diffusivity estimate, yet our limited understanding prevents us from assigning a reasonable value for $\Gamma^T$.

The other inadequacy involves the driving mechanism of mixing. Although turbulent mixing dominates ocean mixing, there are considerable mixing events caused by the release of potential energy due to unstable temperature or salinity stratification (while the density stratification is stable), that is, double diffusion (Schmitt, 1994). Double diffusion has two manifestations, salt finger and diffusive convection. The former is associated with warmer, salter water overlying colder, fresher water; and the latter corresponds to the opposite scenario. Due to their unique requirements of vertical structures for temperature and

salinity, diffusive convection is mostly prominent in the polar and subpolar regions, while salt finger prevails in the tropics and sub-tropical regions (van der Boog et al., 2021); and salt finger is our focus in this study. For the importance of salt finger mixing, analysis of global thermohaline staircase indicated salt finger only contributes a small fraction of the required energy to sustain mixing (van der Boog et al., 2021); however, not all salt finger events present staircases (St. Laurent and Schmitt, 1999), and the regional effects of salt finger mixing can be much profound (Fine et al., 2022). Some studies

suggested salt finger mixing is significant when turbulent mixing is weak, while others suggested salt finger and turbulence can co-exist and interact with each other (Ashin et al., 2023). Unlike turbulent mixing, salt finger mixing, supplied by the release of potential energy, acts to strengthen the density stratification with a negative value of $K_\rho$. With $P$ being negligible, the balance between $B$ and $\varepsilon$ leads to $R_f/(1-R_f)=-1$, and hence $K_\rho=-\varepsilon/N^2$ is applied to salt finger (McDougall, 1988). Therefore, if the mixing mechanism is not identified clearly, the conventional Osborn relation can estimate neither the correct sign nor

the accurate magnitude of eddy diffusivity of density for salt finger mixing. Besides, the eddy diffusivities of heat, salt and density for salt finger mixing are inequivalent, namely $K_\theta<K_S$ (Schmitt et al., 2005). Therefore, $K_\theta$ and $K_S$ for salt finger mixing cannot be estimated by the Osborn relation; and they can be calculated by a different manner involving the dissipation ratio $\Gamma^F$ (note that $\Gamma^F$ for salt finger is equivalent to $-K_\theta/K_\rho$ instead of $R_f/(1-R_f)$; St. Laurent and Schmitt, 1999), density ratio $R_\rho$ (describing the relative contributions of temperature and salt to density) and density flux ratio $r$ (the ratio of

vertical heat flux to vertical salt flux) (see Section 2.3).



To overcome the shortcomings mentioned above, we turn to open microstructure datasets (Section 2), to first identify salt finger mixing from turbulent mixing (Section 3). Then, we explore the variability of $\Gamma^T$ for turbulent mixing (Section 4.1), and examine $\Gamma^F$ and the relation between $R_\rho$ and $r$ for salt finger mixing (Section 4.2). We also derive diffusivities $K_\rho$, $K_\theta$ and $K_S$ and analyze them for both turbulent mixing and salt finger mixing (Section 5). A summary is given in Section 6.

## 2 Data and Methods

### 2.1 Data

We first thank the Climate Process Team for publicly sharing the "Microstructure Database" (MacKinnon et al., 2017). The data used in this study are selected from the shared microstructure sampling projects covering global oceans. Since the calculation of dissipation ratio requires the dissipation rate of thermal variance ($\chi_\theta$), we chose five projects that provide this

variable. Besides $\chi_\theta$, we also use $\varepsilon$, temperature $\theta$ and salinity $S$, which have all been standardized to the same vertical grid for each project. The locations, operating period, etc. of the five projects are given in Table 1 and Fig. 1. The MIXET projects are performed in the western equatorial Pacific, while the BBTRE and NATRE are conducted in the Atlantic between 40°S and 40°N. Salt finger is always active in the mid-to-low latitudes of the Atlantic, while its occurrence in the Pacific shows strong temporal variation (Oyabu et al., 2023). These data provide a great opportunity to investigate the

spatial-temporal variation of dissipation ratio and eddy diffusivity induced by turbulent mixing and salt finger mixing.

**Table 1. Information on the projects used in this study.**

| Project | Location | Period | Profile Number | Vertical Resolution (m) |
|---------|----------|--------|----------------|-------------------------|
| MIXET1 | 156°E, 0°-2°N | 04.20-05.14, 2012 | 51 | 1 |
| MIXET2 | 156°E, 0°-5°N | 10.25-11.18, 2012 | 101 | 1 |
| BBTRE96 | 10°-30°W, 12°-26°S | 01.22-02.27, 1996 | 74 | 0.5 |
| BBTRE97 | 15°-40°W, 10°-26°S | 03.13-04.18, 1997 | 89 | 0.5 |
| NATRE | 20°-30°W, 24°-27°S | 03.25-04.22, 1992 | 150 | 0.5 |

MIXET: MIXing in the Equatorial Thermocline (Waterhouse et al., 2014; Richards et al., 2015)

BBTRE: Brazil Basin Tracer Release Experiment (Polzin et al., 1997)

NATRE: North Atlantic Tracer Release Experiment (St. Laurent and Schmitt, 1999; Polzin and Ferrari, 2004)



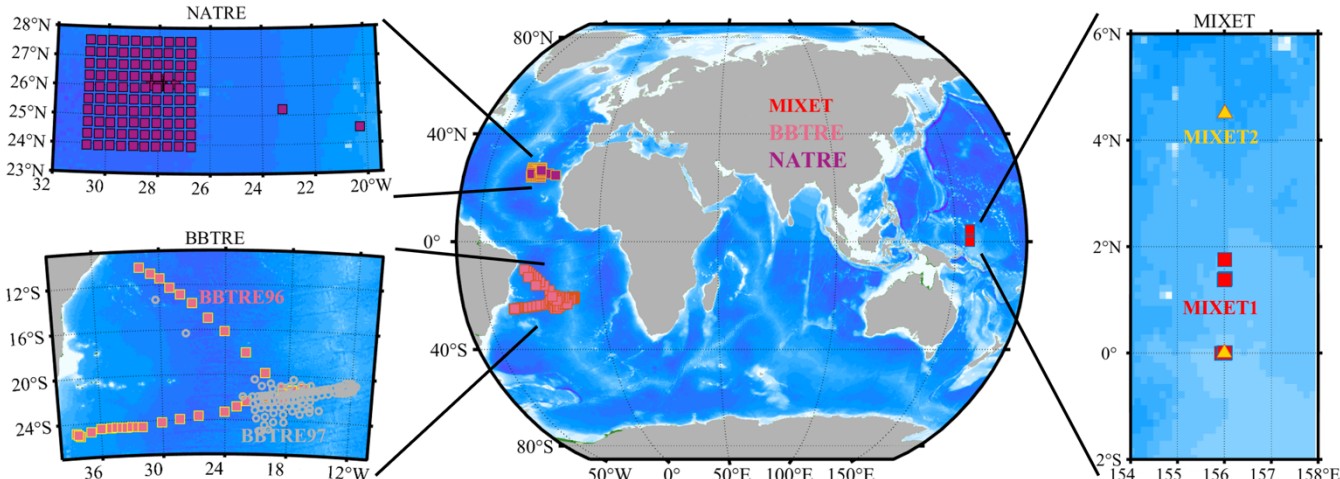

**Figure 1: Station locations of the projects used in this study.**

## 2.2 Identifying turbulent mixing and salt finger mixing

The profiles are divided into half-overlapped patches for further analysis. Following St. Laurent and Schmitt (1999), we choose 10 times of the vertical resolution as patch size, that is, 10 m (5 m) for projects with vertical resolution of about 1 m (0.5 m). We first examine if and which type of double diffusion is favorable for each patch in thermodynamical sense by the Turner angle, $Tu=\text{atan}^{-1}(\alpha\theta_z-\beta S_z,-\alpha\theta_z+\beta S_z)$ (Ruddick, 1983). Here, $\alpha$ and $\beta$ are the thermal expansion and saline contraction coefficients, respectively; $\theta_z$ and $S_z$ are the vertical gradients of the original temperature and salinity profiles, respectively; and "atan$^{-1}$" is the four-quadrant inverse tangent. $Tu$ varies between -180° and 180°, dividing water column into four thermodynamical regimes: doubly stable ($|Tu|<45°$), salt finger favorable ($45°<Tu<90°$), diffusive convection favorable ($-90°<Tu<-45°$), and gravitationally unstable ($|Tu|>90°$) (Ruddick 1983). $Tu$ is related to density ratio $R_\rho$ by $R_\rho=-\tan(Tu+45°)$. We exclude weak double diffusion signals ($45°<Tu<60°$ for salt finger favorable and $-60°<Tu<-45°$ for diffusive convection favorable) for further identification.

Besides the specific thermodynamical precondition, distinct statistical features are presented when double diffusion-induced mixing is dominant. First, $\text{Re}_b$ is found to be no greater than $O(10)$ for active double diffusion (Inoue et al., 2007), and salt finger is rare for $\text{Re}_b$ between 10 and $10^4$ (St. Laurent and Schmitt, 1999). $\text{Re}_b$ is defined as $\text{Re}_b=\varepsilon/\nu N^2$, where $\nu$ is molecular viscosity coefficient. Moreover, double diffusion generally corresponds to elevated $\chi_\theta$ (St. Laurent and Schmitt, 1999; Inoue et al., 2007), and the magnitude of $\chi_\theta$ is significantly larger than $\varepsilon$ when double diffusion prevails and turbulence is absent (Nagai et al., 2015). Therefore, we use $\text{Re}_b<25$ and $|\chi_\theta|/|\varepsilon|\geq7$ as additional criteria for the identification of double diffusion.

For doubly stable and gravitationally unstable water column, since their thermodynamical condition excludes the existence of double diffusion, we assume the mixing within the column is uniquely induced by turbulence only. The most prominent difference between turbulence patches with $|Tu|<45°$ and those with $|Tu|>90°$ is that $\text{Re}_b$ of the former is significantly smaller




than that of the latter. And $|Tu|>90°$ generally means the presence of overturns. Therefore, the former patches are grouped as "weak turbulence", and the latter are "energetic turbulence".

Based on $Tu$, $Re_b$ and $|\chi_\theta|/|\varepsilon|$, we classify the dominant mixing mechanisms into four types: weak turbulence ($|Tu|<45°$ with small $Re_b$), energetic turbulence ($|Tu|>90°$ with large $Re_b$), salt finger ($60°<Tu<90°$, $Re_b<25$ and $|\chi_\theta|/|\varepsilon|\geq7$), and diffusive convection ($-90°<Tu<-60°$, $Re_b<25$ and $|\chi_\theta|/|\varepsilon|\geq7$). Diffusive convection prevails mostly in the polar and subpolar regions (van der Boog et al., 2021); thus, it is rarely identified in this study (Section 3). As a result, diffusive convection is excluded from further analysis.

**2.3 Estimating eddy diffusivities for turbulent mixing and salt finger mixing**

Assuming steady and homogenous state, the production-dissipation balances for TKE (Osborn, 1980) and thermal variance (Osborn and Cox, 1972) are valid for both turbulence and salt finger (St. Laurent and Schmitt, 1999; Inoue et al., 2007),

$$\left(1-R_f\right)K_\rho N^2 - R_f\varepsilon = 0, \tag{1}$$

$$2K_\theta\theta_z^2 - \chi_\theta = 0. \tag{2}$$

Define a general form of dissipation ratio $\Gamma$ as $\frac{\chi_\theta N^2}{2\varepsilon\theta_z^2}$ (Oakey, 1985), combining (1) and (2) yields

$$\Gamma = \left(\frac{R_f}{1-R_f}\right)\frac{K_\theta}{K_\rho} = \left(\frac{R_f}{1-R_f}\right)\left(\frac{R_\rho-1}{R_\rho}\right)\left(\frac{r}{r-1}\right) = \frac{\chi_\theta N^2}{2\varepsilon\theta_z^2} \tag{3}$$

where density flux ratio $r = \frac{\alpha K_\theta\theta_z}{\beta K_S S_z} = \frac{K_\theta}{K_S}\cdot R_\rho$.

For turbulent mixing, $\Gamma^T=R_f/(1-R_f)$; and the eddy diffusivities of heat, salinity and density for turbulent mixing are $K_\theta^T = K_S^T = K_\rho^T = \Gamma^T\frac{\varepsilon}{N^2}$. Here, we use superscripts "T" and "F" to indicate turbulent mixing and salt finger mixing, respectively.

For salt finger mixing, with $R_f/(1-R_f)=-1$, the eddy diffusivity of density can still be derived from the Osborn relation as $K_\rho^F = -\frac{\varepsilon}{N^2}$ (McDougall, 1988), which is five times of the conventional Osborn relation estimate and has a negative sign. However, the eddy diffusivities of heat and salinity for salt finger mixing are more complex (Schmitt et al., 2005),

$$K_\theta^F = \Gamma_\theta^F\frac{\varepsilon}{N^2} = \left(\frac{R_\rho-1}{R_\rho}\right)\left(\frac{r^F}{1-r^F}\right)\frac{\varepsilon}{N^2}, \tag{4}$$

$$K_S^F = \Gamma_S^F\frac{\varepsilon}{N^2} = \frac{R_\rho-1}{1-r^F}\frac{\varepsilon}{N^2}. \tag{5}$$

Note that (4) is actually $K_\theta^F = \chi_\theta/2\theta_z^2$, but in a form analogous to the Osborn relation. And these "analogical" Osborn relations for salt finger indicate the "effective" dissipation ratios for heat and salt are in different forms; but both are deeply related to the density flux ratio $r^F$ and $R_\rho$. $r^F$ can be derived as $R_\rho\frac{\chi_\theta N^2}{2\varepsilon\theta_z^2}/\left(R_\rho\frac{\chi_\theta N^2}{2\varepsilon\theta_z^2}+R_\rho-1\right)$, and then used to infer $K_\theta^F$ and $K_S^F$.



# 3 Statical features of turbulent mixing and salt finger mixing

Figure 2 suggests water properties vary greatly for the five projects, and Table 2 lists the proportions of patches for each
mixing type. For the MIXET projects in the western equatorial Pacific, the *Tu* distribution in spring (MIXET1) shows a
distinct shape from the autumn one (MIXET2). In spring, *Tu* shows double peaks at -30° and 110°, suggesting mixing is
alternately dominated by weak and energetic turbulence, although the salt finger contribution accounts for 4.1% of the total
patches and cannot be neglected. However, the autumn distribution is obviously unimodal, peaking at ~45°; and the
dominant mixing types are first weak turbulence and secondly salt finger (~51.5% and 11.3%, respectively), with negligible
energetic turbulence and diffusive convection. For the BBTRE projects, although they are conducted at different years, the
operating seasons are similar: one in late-summer and the other early-autumn (Southern Hemisphere), so the seasonal
variation cannot be studied. Their *Tu* distributions are similarly bimodal, with a leading peak at 70° and a weak one at -40°,
suggesting the waters are mostly salt finger-favorable (although only about 5.9% is confirmed to be salt finger) and stable
(33.3%), with rare energetic turbulence and neglectable diffusive convection-favorable contribution. For the NATRE, salt
finger overwhelms the others, occupying more than 21% of the total patches; weak and energetic turbulence together hold
13.3%, with the diffusive convection favorable still being negligible (1%). For these five projects, although almost half of
the patches are salt finger favorable, only 9.7% of them shows clear salt finger features. Weak turbulence has a higher
percentage (32.0%), followed by 6.6% of energetic turbulence. Diffusive convection occurs less than 0.5% of the total
patches, and is therefore negligible.

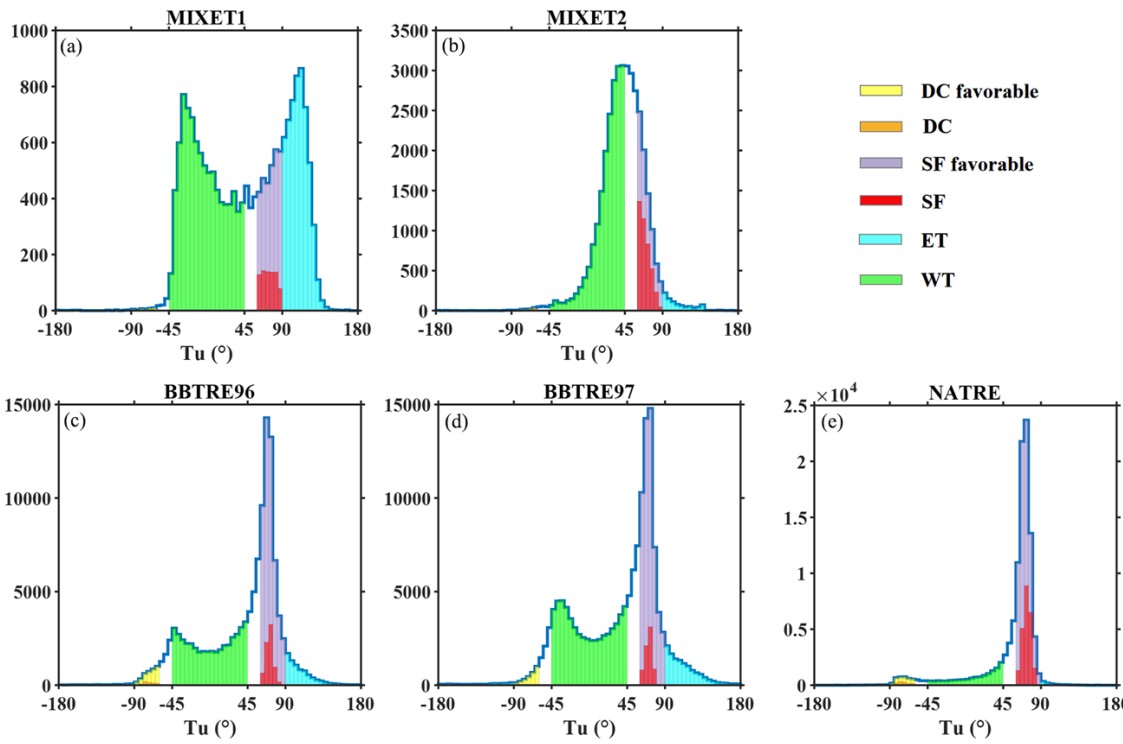





**Figure 2: Histograms of patch-averaged *Tu* for different projects. Different *Tu* ranges of mixing types are marked by different colors: yellow for diffusive convection favorable (DC favorable; -90°<*Tu*<-60°), light purple for salt finger favorable (SF favorable; 60°<*Tu*<90°), cyan for energetic turbulence (ET), and green for weak turbulence (WT). The red and orange bars denote the actual patch numbers of salt finger (SF) and diffusive convection (DC) selected by two more criteria, $Re_b$<25 and $|\chi_\theta|/|\varepsilon|\geq5$, respectively.**

**Table. 2. Proportions of patches with energetic turbulence, weak turbulence, salt finger, and diffusive convection to the total patch number for each project, and the sums for all the projects.**

|  | Proportion (%) | | | |
|---|---|---|---|---|
|  | energetic turbulence | weak turbulence | salt finger | diffusive convection |
| MIXET1 | 29.56 | 47.05 | 4.11 | 0.06 |
| MIXET2 | 2.48 | 51.48 | 11.32 | 0.16 |
| BBTRE96 | 6.55 | 33.31 | 5.91 | 0.53 |
| BBTRE97 | 8.67 | 38.19 | 4.56 | 0.08 |
| NATRE | 1.10 | 12.21 | 21.95 | 1.09 |
| All | 6.60 | 32.00 | 9.70 | 0.46 |

We compare the statistical differences of $Re_b$, $\varepsilon$, $N^2$, and $\chi_\theta$ for energetic turbulence, weak turbulence and salt finger by considering all the patches from the five projects (Fig. 3). The salt finger patches are featured with the weakest turbulence

intensity compared with weak and energetic turbulence patches, whose median $Re_b$ are 5.0, 18.2 and 132.7, respectively. The median $Re_b$ of energetic turbulence is slightly smaller than that reported in Mashayek et al. (2017) but close to the result of Ijichi and Hibiya (2018). Since the samples given here are from five different projects, their $Re_b$ distributions are actually different: For MIXET projects, the median $Re_b$ of energetic turbulence is small, only about 50; while the rest projects generally have a median $Re_b$ around 200 for energetic turbulence. The variations of $\varepsilon$ for different mixing types differ little,

mostly ranging from $3\times10^{-12}$ to $3\times10^{-8}$ W kg$^{-1}$. Although the median $\varepsilon$ for energetic turbulence is not obviously different from those for weak turbulence and salt finger ($7.8\times10^{-11}$, $7.9\times10^{-11}$ and $1.1\times10^{-10}$ W kg$^{-1}$, respectively), it should be noted that most large $\varepsilon$ values are induced by energetic turbulence. Distributions of $\chi_\theta$ of weak turbulence and energetic turbulence differ little, but $\chi_\theta$ of salt finger is clearly greater, in terms of variation ranges (salt finger: $3\times10^{-11}$-$10^{-7}$ °C$^2$ s$^{-1}$; weak turbulence and energetic turbulence: $10^{-13}$-$10^{-7}$ °C$^2$ s$^{-1}$) and median values (salt finger: $1.8\times10^{-9}$ °C$^2$ s$^{-1}$; energetic turbulence

and weak turbulence: $1.5\times10^{-11}$ °C$^2$ s$^{-1}$). Earlier studies considered the doubly stable regime as no mixing or excluded it from analysis (Inoue et al., 2007); however, besides some slight differences of proportion in large $\chi_\theta$ and $\varepsilon$, energetic turbulence and weak turbulence share very similar distributions of $\chi_\theta$ and $\varepsilon$ (Figs. 3b, d), suggesting the doubly stable regime does not





mean an absence of turbulence and should be dominated by weak turbulence. Stratification also presents different features for different mixing types. The energetic turbulence has the weakest stratification with a median of $6.1 \times 10^{-7}$ s$^{-2}$, only 1/5 of

that for weak turbulence. And salt finger presents the strongest stratification ($1.9 \times 10^{-5}$ s$^{-2}$). Clearly, the identified patches with energetic turbulence, weak turbulence and salt finger have distinct turbulent features, verifying the validity of the chosen criteria.

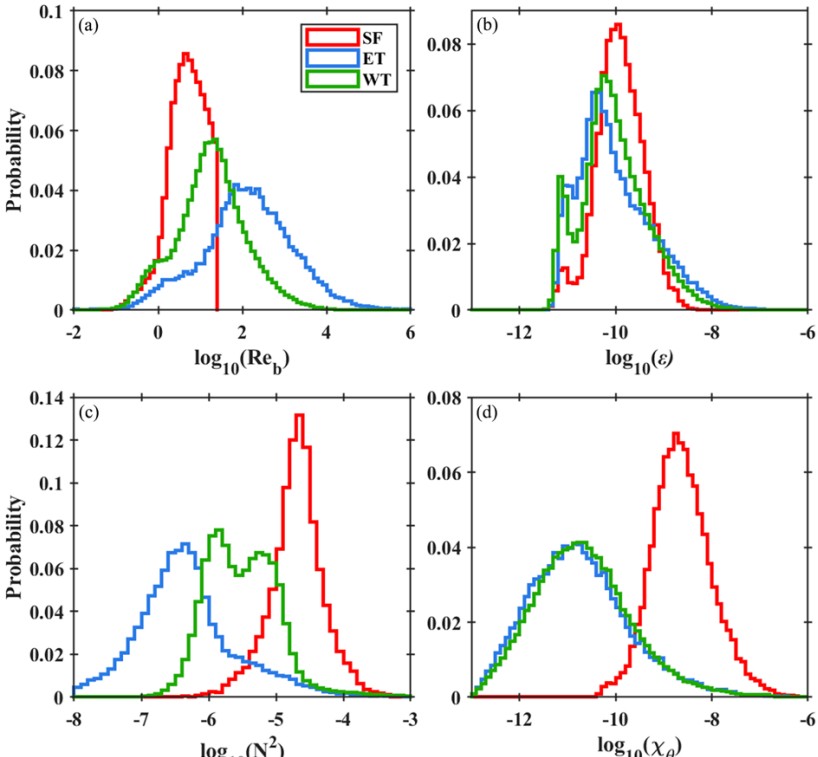

**Figure 3: Probability-normalized histograms of $\log_{10}(Re_b)$ (a), $\log_{10}(\varepsilon)$ (b), $\log_{10}(N^2)$ (c), and $\log_{10}(\chi_\theta)$ (d) for different mixing types:**
**SF (salt finger), ET (energetic turbulence) and WT (weak turbulence). Data of the five projects are taken as the whole collection.**

A normalized occurrence frequency is calculated to quantify the vertical variation of each mixing type (Fig. 4). Taking energetic turbulence as an example, we first divide energetic turbulence patch number of each depth bin to the total energetic turbulence patch number of the whole project; then, to eliminate the vertical variation of observation frequency, we divide the results by the total patch number within the same depth bin. This occurrence frequency is eventually normalized between

0 and 1 using its maximum. Consistent with some observations in the upper thermocline (Schmitt et al., 2005; van der Boog et al., 2021), salt finger is mostly prevailing in the upper 500-1000 m for all projects, with their occurrence frequencies reaching 1. For the MIXET projects and NATRE, the occurrence frequencies of salt finger gradually become weak and near zero with depth increasing to the seafloor. However, for the BBTRE projects, salt finger sharply disappears between 1000 and 2000 m and re-occurs at deeper depth (see Figure 10). The depth-colored T-S diagrams suggest the vertical transition of




different water masses is responsible for the sudden disappearing of salt finger (Fig. 5). It is clear to see that both $\theta$ and $S$ decrease with depth in most water columns, providing the basic precondition for salt finger. However, this tendency changes obviously between 1000 and 2000 m. At this depth range, $\theta$ changes little, but $S$ increases drastically by at least 0.5; this prevents the occurrence of salt finger. This depth is just where the fresher Antarctic Intermediate Water transits to the North Atlantic Deep Water. Consequently, the occurrence frequency of salt finger is severely weakened at this depth. On the

contrary to salt finger, energetic turbulence generally becomes more prevailing with increasing depth for most projects. The remarkably weak background stratification may contribute a lot to the flourish of energetic turbulence at depth, where even a weak perturbation can fully develop.

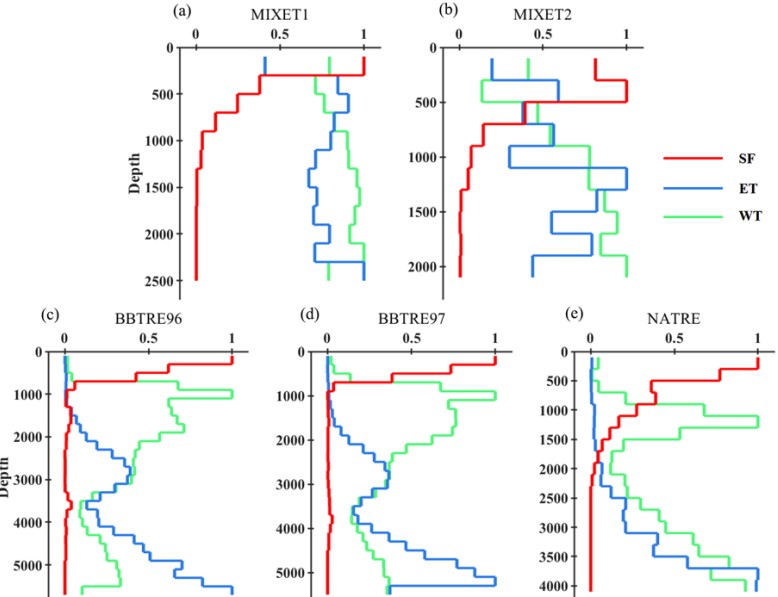

**Fig. 4. Vertical variations of normalized occurrence frequency of salt finger (SF), energetic turbulence (ET) and weak turbulence**

**(WT) for the five projects. The depth range is from 100 m to the deepest measurements, with a bin size of 200 m.**

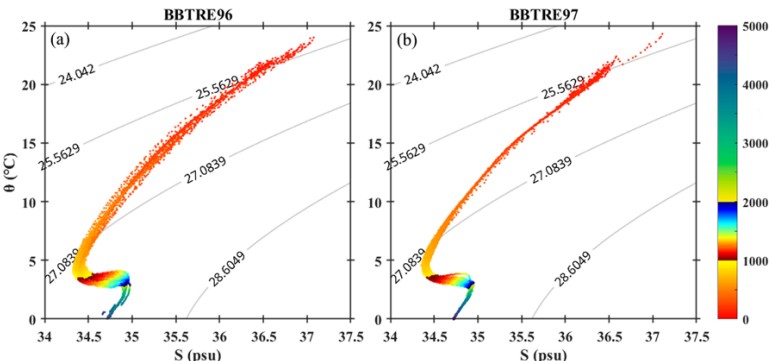

**Fig. 5. T-S diagrams for BBTRE96 and BBTRE97. Color indicates patch depth, and the contour indicates isopycnic.**





# 4 Γ variation of turbulence and salt finger

## 4.1 Γ variation of turbulence

We explore the variation of $\Gamma^T$ first. Figure 6 suggests $\Gamma^T$ varies in distinct manners for different projects. Results for the MIXET projects suggest $\Gamma^T$ in the western equatorial Pacific is significantly seasonally variable. In spring (MIXET1), $\Gamma^T$ of energetic turbulence varies between $2.5 \times 10^{-2}$ and 1.7 ($10^{th}$-$90^{th}$ percentiles) with a median of 0.23, smaller than that of weak turbulence ranging between $1.4 \times 10^{-1}$ and 2.8 and peaking at 0.52. $\Gamma^T$ in autumn is significantly elevated (MIXET2), and the medians and variation ranges for energetic turbulence and weak turbulence are [0.41 and from $3.4 \times 10^{-2}$ to 8.8] and [0.58 and

from $1.7 \times 10^{-1}$ to 2.3], respectively. For the BBTRE projects, $\Gamma^T$ of weak turbulence varies little between different years, with most patches varying between $10^{-2}$ and 10, although the median value in 1997 (0.35) was greater than that in 1996 (0.20). $\Gamma^T$ of energetic turbulence is larger in 1997 than that in 1996, with median values of 0.48 and 0.20, respectively. Estimates from the NATRE also suggest $\Gamma^T$ largely scatters between $10^{-2}$ and 10 for most patches; their median $\Gamma^T$ values are 0.71 and 0.33 for energetic turbulence, and are 0.41 and 0.50 for weak turbulence. To summarize, besides the BBTRE and energetic

turbulence of the MIXET projects showing a median value close to 0.2, the rest estimates are all clearly greater than 0.2. $\Gamma^T$ for the five projects mostly vary within three orders of magnitude from $10^{-2}$ to 10, in line with other observations (Ijichi and Hibiya, 2018; Vladoiu et al., 2021; Li et al., 2023).

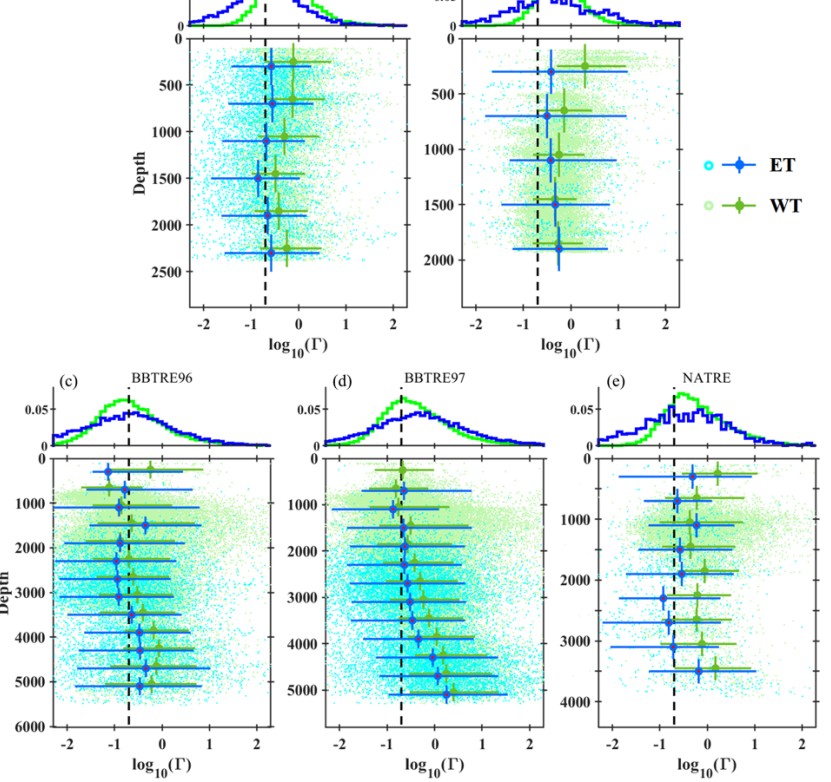





**Fig. 6. Variations of $\Gamma^T$ of energetic turbulence (ET) and weak turbulence (WT). Each panel consists of two sub-panels, with the**
**upper one showing probability-normalized histogram of $\Gamma^T$, and the lower one being $\Gamma^T$-depth scatters; the median value of each**
**depth bin is marked by a larger, darker dot overlying a cross marker, with horizontal bar indicating the 10th to 90th percentile**
**range and vertical bar indicating the depth-bin range. The median $Re_b$ values are compared between energetic turbulence and**
**weak turbulence at each depth bin, and the median $\Gamma^T$ corresponding to the larger $Re_b$ is marked by a red dot. The conventional**
**value of $\Gamma^T$, namely 0.2, is represented by the dashed black line.**

For different projects, $\Gamma^T$ varies with depth in different way. For the MIXET1, $\Gamma^T$ of both energetic turbulence and weak turbulence fluctuate around their statistical median values weakly. For the MIXET2, the depth-median $\Gamma^T$ of energetic turbulence varies between 0.2 and 0.7 alternately, with a slightly increasing trend. However, $\Gamma^T$ of weak turbulence shows a clear decreasing from 2.5 at 300 m to 0.6 at 1400 m; then, it slightly increases to 0.8 at 1900 m. The $\Gamma^T$ of weak turbulence for the BBTRE96 fluctuates around 0.2 in the upper 300 m, then it increases to ~1 at 4400 m and then decreases to ~0.6 at 5200 m. The scenario for energetic turbulence shares a similar picture. $\Gamma^T$ of weak turbulence for the BBTRE97 departs little from 0.2 at depths above 1800 m, then monotonically increasing to ~2.3 at the deepest depth around 5200 m. $\Gamma^T$ of energetic turbulence varies in a similar way in vertical, except the depth where trends change is 3000 m. For NATRE, $\Gamma^T$ of energetic turbulence firstly decreases from 0.6 to 0.1 at 2300 m, then increases to 0.8 at 3500 m. As for weak turbulence, $\Gamma^T$ stays around 0.8 between 600 and 3000 m and then increases beyond unity at 3500 m. In term of general trend by linear fitting $\Gamma^T$ with depth, the five projects show two distinct vertical patterns of $\Gamma^T$: One is the vertically decreasing pattern represented by the MIXET projects, and the other is the vertically increasing one suggested by the rest projects over the midlatitude of the Atlantic. Vertically increasing $\Gamma^T$ was also reported by Ijichi and Hibiya (2018). Their data collection sites spread over mid-to-high latitudes of the Pacific and Southern Ocean. $\Gamma^T$ also presented a clear vertically increasing trend in the upper 500 m of the South China Sea north of 10°N (Li et al., 2023). Combining all these observational results, we suggest $\Gamma^T$ in the equatorial area should be treated differently, since it may decrease in the vertical, contrary to the vertically increasing trend away from the equator.

The full-depth statistics of the five projects disagree about which $\Gamma^T$ is larger for energetic turbulence and weak turbulence. However, when comparing $\Gamma^T$ values of energetic turbulence and weak turbulence in the same depth bin, $\Gamma^T$ of energetic turbulence is mostly smaller than that of weak turbulence. Considering that $Re_b$ is reported to deeply modulate the variation of $\Gamma^T$ (Mashayek et al., 2017; Monismith et al., 2018), and that energetic turbulence and weak turbulence have clearly different $Re_b$ distributions (Fig. 3), we found energetic turbulence with smaller $\Gamma^T$ generally has larger $Re_b$ than weak turbulence, indicating a negative correlation between $\Gamma^T$ and $Re_b$.

We then investigate the relations between $\Gamma^T$ and $Re_b$ for energetic turbulence and weak turbulence (Fig. 7). For the MIXET1, $\Gamma^T$ of weak turbulence first decreases from 3.5 to 0.5 with $Re_b$ increasing from 0.1 to 1, suggesting a relation of $\Gamma^T \propto Re_b^{-1}$, and then it weakly increases to 0.7 with $Re_b$ reaching 100; and a weak decreasing in line with $\Gamma^T \propto Re_b^{-1/2}$ can be observed for $Re_b>100$. For energetic turbulence, $\Gamma^T$ generally decreases with $Re_b$, indicating $\Gamma^T \propto Re_b^{-1/2}$; this relation is consistent with the



observations in the western Mediterranean Sea (Vladoiu et al., 2021). The pattern for the MIXET2 is similar to that for the MIXET1, although $\Gamma^T$ of weak turbulence decreases in a smaller rate when $Re_b$ is small and indicates $\Gamma^T \propto Re_b^{-1/2}$. Excluding the bins with few data points, $\Gamma^T$ of weak turbulence for the BBTRE96 shows a weak increasing trend from 0.2 to 0.3 as $Re_b$

grows from 10 to $10^3$ and a weak decreasing trend with $Re_b$ exceeding $10^3$. $\Gamma^T$ and $Re_b$ of weak turbulence for the BBTRE97 show similar relationships as those for the BBTRE96. The weak turbulence trends for the BBTRES are the same as the estimates reported by Ijichi et al. (2020); and its shape is similar to the upper bound of the nonmonotonic $\Gamma^T \sim Re_b$ relation proposed by Mashayek et al. (2017). It is notable that the scenario for energetic turbulence is distinct; $\Gamma^T$ generally decreases from 5 to less than 0.1 with $Re_b$ between 10 and $2.5 \times 10^4$ for the BBTRE96, forming a fitting slope steeper than -1/2 but

flatter than -1. $\Gamma^T$ of energetic turbulence for the BBTRE97 also shows a similar decreasing trend with $Re_b$. Except for the bins with few samples when $Re_b < 1$ and $Re_b > 10^4$, $\Gamma^T$ of weak turbulence for the NATRE generally increases from 0.5 to 0.7, while $\Gamma^T$ of energetic turbulence monotonically decreases from ~1 at $Re_b = 10^2$ to ~0.1 at $Re_b = 10^4$, suggesting $\Gamma^T \propto Re_b^{-1/2}$.

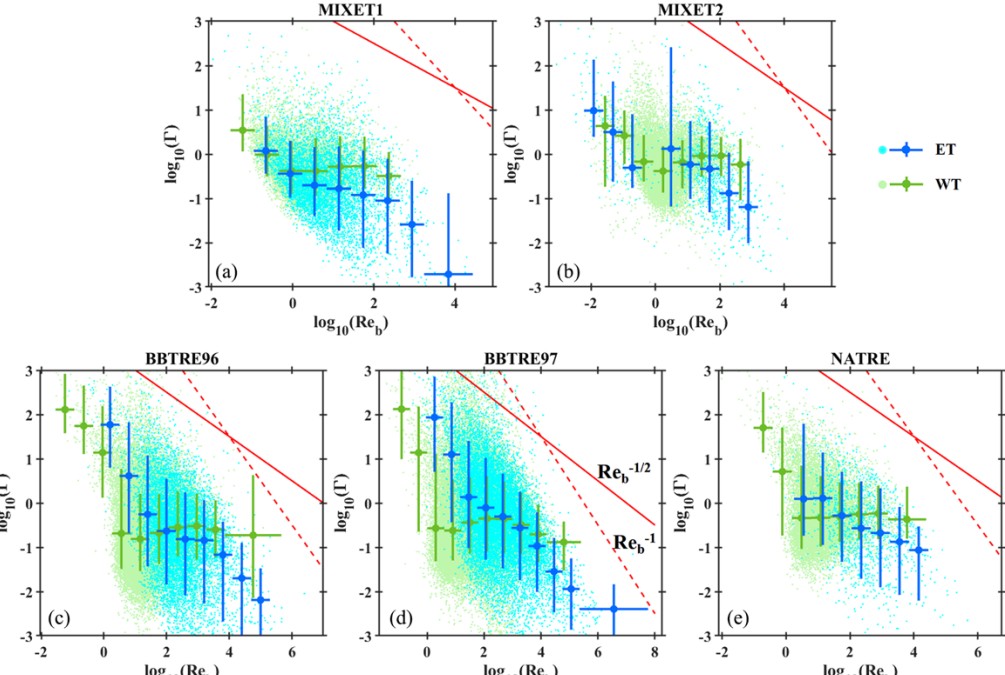

**Fig. 7. Relations between $\Gamma^T$ and $Re_b$ for energetic turbulence (ET) and weak turbulence (WT). Overlying the light-color scatters**

**of individual patches, $Re_b$ -binned median values are marked by large darker dots; and the bin size and the 10th-90th percentile range of $\Gamma^T$ are denoted by the horizontal and vertical bars, respectively. The solid and dashed red lines mark $\Gamma^T \propto Re_b^{-1/2}$ and $\Gamma^T \propto Re_b^{-1}$, respectively.**

Although $\Gamma^T$ generally decreases with $Re_b$ in most cases of the five projects, the decreasing rate varies with projects and $Re_b$ ranges. There are several cases showing $\Gamma^T$ stays constant or even increases with $Re_b$. These suggest $\Gamma^T$ is not solely

modulated by $Re_b$; and there may be other factors that influence $\Gamma^T$ in a comparable or even dominating role relative to $Re_b$.



$R_{OT}$ is reported as such a parameter that regulates $\Gamma^T$ more strongly than $Re_b$, $\Gamma^T \propto R_{OT}^{-4/3}$ (Ijichi and Hibiya, 2018). $R_{OT}$ is the ratio of the Ozmidov scale $L_O$ to the Thorpe scale $L_T$, $R_{OT}=L_O/L_T$ with $L_O=\varepsilon^{1/2}/N^{3/2}$ and $L_T = \langle \delta_T^2 \rangle^{\frac{1}{2}}$, where the Thorpe displacement $\delta_T$ is the depth difference of a water parcel between the original and sorted potential temperature profiles of an overturn. Overturns are identified by the cumulative Thorpe displacement $\sum \delta_T$ (Mater et al., 2015; Ijichi and Hibiya, 2018).

Because the vertical resolution of temperature profiles is 1 m or 0.5 m, overturns with vertical size of $O(1)$ m or smaller cannot be identified. Additionally, the identified overturns with size smaller than 10 m or greater than 400 m are excluded from analysis, because the former contain too few data points and the latter are possibly the vertical structures of different water masses instead of genuine turbulent overturns. We also estimate the overturn-averaged $Tu$, $\Gamma^T$ and $Re_b$. Due to the coarse vertical resolution of temperature profiles used in our study, only a few overturns meet the identification criteria for

each project; as a result, the overturns of the five projects are taken as one collection (total overturn number is 3862).

Figure 8 shows the overturn-based relation between $\Gamma^T$ and $R_{OT}$. Since most overturns are identified at depth, with only one fifth shallower than 1000 m but more than one third at depth below 2000 m, the overturn-based $\Gamma^T$ is clearly greater than 0.2, with a median value of 0.91. In Fig. 8, although overturns are evenly scattered in the $R_{OT}$-$\Gamma^T$ space, the probability density shows they concentrate around two sites mostly, one with $R_{OT}$ and $\Gamma^T$ of (0.03, 1.19) and the other (0.56, 0.53). These two

clusters are well distinguished by $Re_b$, with the first location corresponding to $Re_b<160$ (median value is 25) and the other to $Re_b>160$ (median value is 835). For both clusters, the contours of probability density tilt at slopes of -4/3, confirming $\Gamma^T \propto R_{OT}^{-4/3}$ is valid for each cluster. However, the general trend between $R_{OT}$ and $\Gamma^T$ for the whole data collection is much flatter, with a slope of only about -1/2. Comparing $Re_b$ of the two clusters, it is easy to find that $Re_b$ grows exponentially with $R_{OT}$. Therefore, the general variation of $\Gamma^T$ with the growth of $R_{OT}$ is not only influenced by $R_{OT}$, but also partly affected by $Re_b$.

Supposing $\Gamma^T$ is mostly modulated by these two parameters, and considering the decrease trend of $\Gamma^T$ with $R_{OT}$ is significantly weakened by $Re_b$, this suggests a positively relation between $\Gamma^T$ and $Re_b$.

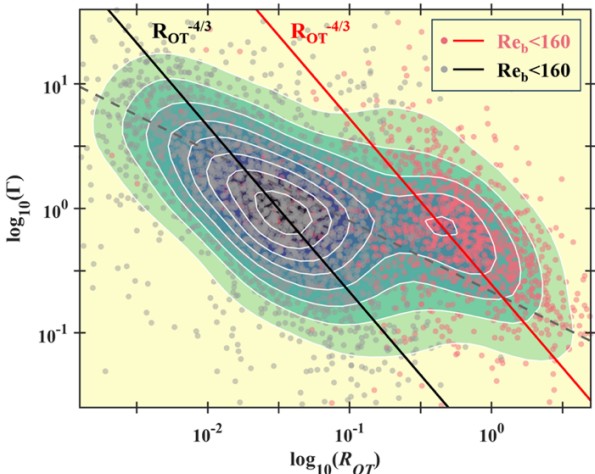

**Fig. 8. Relation between overturn-based $\Gamma^T$ and $R_{OT}$, overturns from the five projects are considered. The shading describes the distribution of probability density, with yellow indicating minimum probability density and blue representing maximum one. The**





overturns are correspondingly divided into two clusters: the gray dots have Re$_b$<160, and the pink ones, Re$_b$>160. The black and red lines represent $\Gamma^T \propto R_{OT}^{-4/3}$, crossing the centers of the two clusters. The gray dashed line is the general relation between $\Gamma^T$ and $R_{OT}$ of the whole data collection.

Figure 9 shows the variation of median value of $\Gamma^T$ jointly binned by Re$_b$ and $R_{OT}$. Note that most parts of the Re$_b$-$R_{OT}$ space are null, with all the data gathered around a band originating from large Re$_b$ and $R_{OT}$ to small Re$_b$ and $R_{OT}$. This confirms that

Re$_b$ and $R_{OT}$ are positively correlated in general. As for the median $\Gamma^T$, although its value is scattered, its general pattern indicates $\Gamma^T$ grows fastest along a direction from small Re$_b$ and large $R_{OT}$ to large Re$_b$ and small $R_{OT}$, suggesting that $\Gamma^T$ is indeed positively correlated with Re$_b$ and negatively correlated with $R_{OT}$. Assuming $\Gamma^T \propto R_{OT}^{-4/3} \cdot Re_b^c$, we substitute the median values of $\Gamma^T$, Re$_b$ and $R_{OT}$ in Fig. 9 into this relation to fit the exponent $c$. The fitting results suggest $c \approx 1/2$ and a relation of $\Gamma^T \approx 10^{-3} \cdot R_{OT}^{-4/3} \cdot Re_b^{1/2}$. The isolines of this relation are shown in Fig. 9, which can well capture the main variation

trend of $\Gamma^T$ with Re$_b$ and $R_{OT}$. Based on the microstructure measurements collected from the upper layer in the South China Sea, Li et al. (2023) presented a relation of $\Gamma^T \approx a R_{OT}^{-4/3} \cdot Re_b^{1/2}$, but $a$ is around 0.02 in that region, one magnitude larger than the value presented here. This is because Re$_b$ have much smaller magnitude in the upper South China Sea, with most Re$_b$ varying between $10^{-1}$ and $10^3$. Therefore, compared with the results in Li et al. (2023), the larger Re$_b$ in this study lead to a relatively smaller $a$. On the other hand, the significant variation of $a$ may suggest some other parameters can influence $\Gamma^T$

besides Re$_b$ and $R_{OT}$.

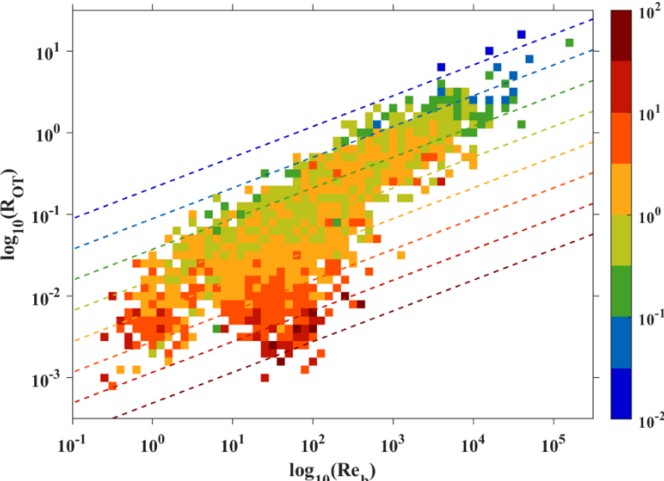

**Fig. 9. Variation of median $\Gamma^T$ binned by $R_{OT}$ and Re$_b$, based on overturn estimates of the five projects. The colored dashed lines indicate the isolines of $10^{-3} \cdot R_{OT}^{-4/3} \cdot Re_b^{1/2}$.**

**4.2 $\Gamma$ variation of salt finger**

$\Gamma^F$ has been widely used to distinguish salt finger from turbulence, since its value is reported to be larger than the conventional $\Gamma^T$ value of 0.2 (St. Laurent and Schmitt, 1999). However, the full-depth observations presented in either this study or previous ones indicate 0.2 is an underestimate of $\Gamma^T$, the difference of dissipation ratio between turbulent mixing





and salt finger mixing in the deep water needs to be examined. Figure 10 presents the variations of $\Gamma_\theta^F$ and $\Gamma_S^F$ with depth. Compared with $\Gamma^T$ varying over three orders of magnitude, both $\Gamma_\theta^F$ and $\Gamma_S^F$ are less variable and change by two orders in

magnitude or as small as one order. The median $\Gamma_\theta^F$ for all samples from the five projects is 0.47, slightly smaller than the $\Gamma^F$ observed in the diurnal thermocline of the Arabian Sea (0.65; Ashin et al., 2023), in the Kuroshio Extension Front (~1; Nagai et al., 2015), and in the thermocline of the western tropical Atlantic (~1.2; Schmitt et al., 2005). The median $\Gamma_\theta^F$ for the five projects are distinct: 0.25, 0.29 and 0.28 for the MIXET1, BBTRE96 and BBTRE97, similar to the conventional $\Gamma^T$ value of 0.2; 0.52 for NATRE, distinguishable from 0.2 but close to their observed $\Gamma^T$ (Fig. 6); 0.98 for the MIXET2, significantly

larger than 0.2 and different from their observed $\Gamma^T$ (Fig. 6). This suggests the dissipation ratio difference between turbulence and salt finger is complex.

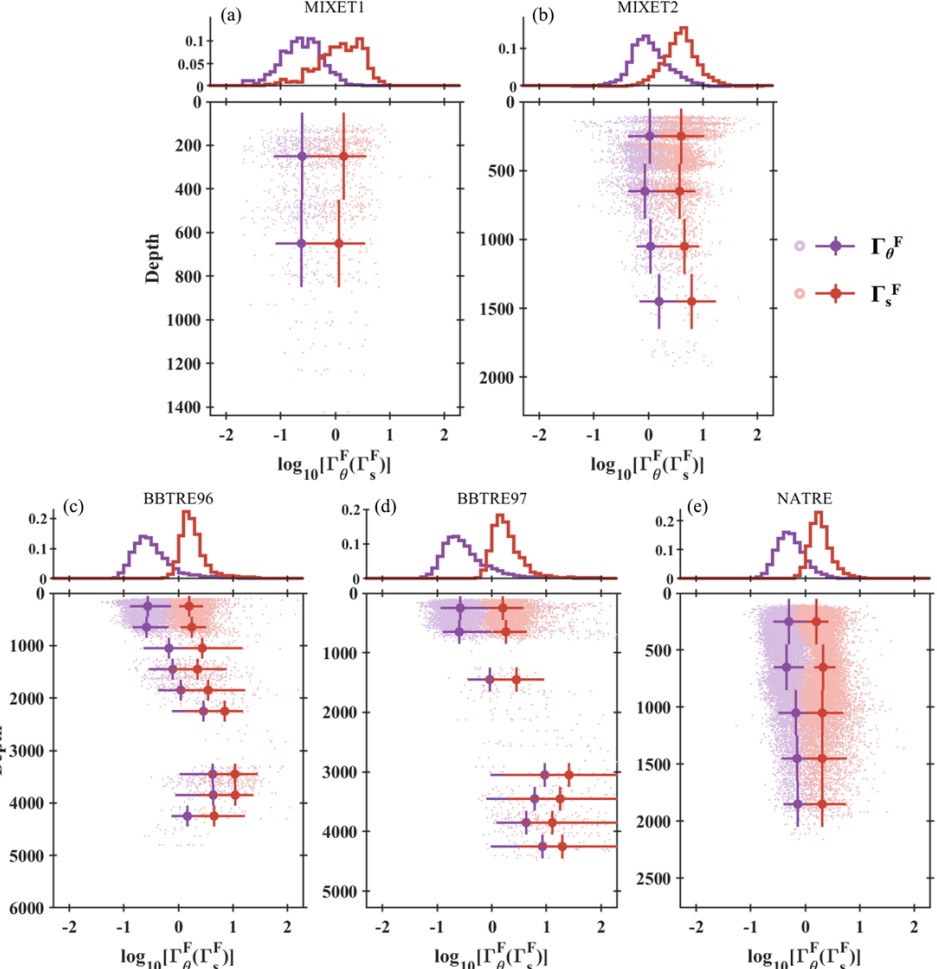

**Fig. 10. Variations of $\Gamma_\theta^F(\Gamma_S^F)$ of salt finger for the five projects. Each panel consists of two sub-panels, with the upper one showing the probability-normalized histograms of $\Gamma_\theta^F$ and $\Gamma_S^F$, and the lower one being their vertical variations. The median value of each**

**depth bin is marked by a larger, darker dot overlying a cross marker, with horizontal bar indicating the 10th to 90th percentile range and vertical bar indicating the depth-bin range.**



Vertically, $\Gamma_\theta^F$ for MIXET1 keeps nearly constant as 0.25. While $\Gamma_\theta^F$ for MIXET2 first decreases from 1 to 0.7 in the upper 700 m and then slightly increases to 2 at 1500 m. $\Gamma_\theta^F$ present a similar vertical trend for both BBTRE96 and BBTRE97: $\Gamma_\theta^F$ is small and stays as a constant within the upper 800 m, with a median of 0.28; with depth increasing to 3000 m, it significantly increases over orders of magnitude, and the median value reaching ~10; it is weakened at deeper depth. Note that the relatively small median $\Gamma_\theta^F$ for the BBTRE projects is mainly caused by the dominant patches with small $\Gamma_\theta^F$ values in the upper 800 m; and $\Gamma_\theta^F$ at depth is actually very large and significantly greater than 0.2 or the observed $\Gamma^T$. The scenario for the NATRE is similar to that for the MIXET2, whose depth-median $\Gamma_\theta^F$ remains nearly consistent around ~0.5, although a very weak increasing trend exists.

The "effective" salt dissipation ratio $\Gamma_S^F$ tends to be obviously larger than $\Gamma_\theta^F$ (Fig. 10). With the overall median $\Gamma_S^F$ of 1.87, the median values of $\Gamma_S^F$ for the five projects are 1.35 (MIXET1), 3.98 (MIXET2), 1.67 (BBTRE96), 1.71 (BBTRE97), and 1.83 (NATRE), floating around the value reported in the thermocline of the western tropical Atlantic of ~2.8 (ref). $\Gamma_S^F$ is strongly positively proportional to $\Gamma_\theta^F$, with the median values of $\Gamma_S^F/\Gamma_\theta^F$ for the five projects being 5.1, 3.7, 6.3, 6.9, and 3.8, respectively. Thus, a general relation of $\Gamma_S^F \approx 5\Gamma_\theta^F$ can be inferred. Due to this correlation, $\Gamma_S^F$ presents vary similar vertical variation as $\Gamma_\theta^F$.

Note that $\Gamma_S^F/\Gamma_\theta^F$ is equivalent to $R_\rho/r^F$. Since $R_\rho$ is relatively easy to calculate, as a result, it is an alternate way to infer the hard-to-measure $r^F$. $R_\rho$ and $r^F$ are the key parameters to estimate the dissipation ratios of heat and salt for salt finger (Section 2.3). Therefore, many studies tried to explore the relation of $R_\rho$ and $r^F$ based on theoretical derivations, laboratory experiments and numerical simulations (Kelley, 1986; Kunze, 1987; Radko and Smith, 2012). Here, the $R_\rho$-$r^F$ diagram colored by probability density for the five projects indicate the salt finger patches are rather scattered (Fig. 11). However, the median $r^F$ binned by $R_\rho$ shows a clear nonmonotonic variability. For $R_\rho$ increasing from 1 to 2.4, $r^F$ decreases from ~0.8 to 0.4; then, it gradually increases to 0.55 with $R_\rho$ approaching 3.7. This correlation between $R_\rho$ and $r^F$ can be well fitted by

$$r^F = \frac{0.79 \cdot R_\rho^2 - 2.96 \cdot R_\rho + 3.18}{R_\rho^2 - 3.26 \cdot R_\rho + 3.46} \tag{6}$$

Compared with other correlation curves (Kelley, 1986; Kunze, 1987; Radko and Smith, 2012), all of them present a $r^F$ decreasing trend for $R_\rho$ smaller than 2, although the variation range and rate differ. The most obvious discrepancy between them is that $r^F$ tends to regain a larger value with $R_\rho$ exceeding 2.4 in our study, while all the other curves decrease little to asymptote to a constant value. The observational result presented here falls in the area outlined by the existing results. For our results, the salt finger patches with $R_\rho < 2.5$ are abundant and mostly concentrated to indicate a negative correlation between $R_\rho$ and $r^F$. It needs to be mentioned that patches with $R_\rho > 2.5$ are much rare and sparsely distributed, making the increasing trend of $r^F$ in larger $R_\rho$ range need to be treated carefully.





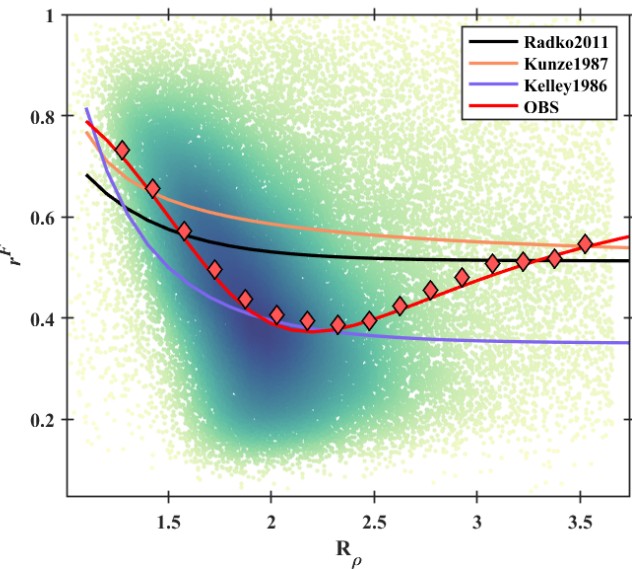

**Fig. 11. Relation between $R_\rho$ and $r^F$. Salt finger patches from all the five projects are considered. Dots are colored by probability density, with darker color indicating larger probability density. The median $r^F$ binned by $R_\rho$ are marked by red diamonds with black edge, and the red curve is the fitting curve. The black, orange and purple curves are adopted from Radko and Smith (2011),**
**Kunze (1987) and Kelley (1986), respectively.**

We also investigate relation between observed $\Gamma_\theta^F$ and $Re_b$ (Fig. 12), which differs considerably between different projects. For the MIXET1, a nearly linear decreasing trend of $\Gamma_\theta^F$ (in logarithmic scale) from ~1 to ~0.1 can be easily observed for all patches with $Re_b$ between 0.3 and 25, indicating $\Gamma_\theta^F \propto Re_b^{-1/2}$. $\Gamma_\theta^F$ for MIXET2 with $Re_b<2.5$ are also well fitted as $\Gamma_\theta^F \propto Re_b^{-1/2}$, but $\Gamma_\theta^F$ for $Re_b>2.5$ tends to remain a constant of 0.7. For the BBTRE projects, when $Re_b<3$, $\Gamma_\theta^F$ decreases at a larger rate
than the MIXET projects, $\Gamma_\theta^F \propto Re_b^{-1}$, and $\Gamma_\theta^F$ stays almost unchanged when $Re_b$ exceeds 3. $\Gamma_\theta^F$ for the NATRE stays as a constant of 0.7 with most $Re_b$ ranging from 1 to 25. Due to the strong correlation between $\Gamma_S^F$ and $\Gamma_\theta^F$, the dependence of $\Gamma_S^F$ on $Re_b$ is similar to that of $\Gamma_\theta^F$, although variation rates are different for some projects. Taking all the projects together, $\Gamma_\theta^F$ and $\Gamma_S^F$ decrease with $Re_b$ in general; however, the decreasing rate varies greatly with projects and different $Re_b$ bands, indicating $\Gamma_\theta^F$ and $\Gamma_S^F$ may also be modulated by variables other than $Re_b$.





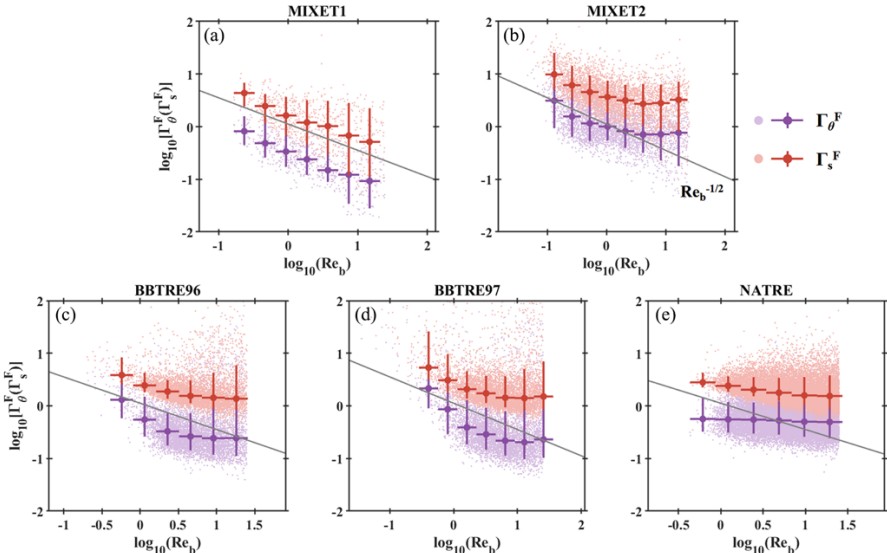


**Fig. 12. Relations between $\Gamma_\theta^F(\Gamma_S^F)$ and $Re_b$ for the five projects. Overlying the light-color scatters of individual patches, the $Re_b$-binned median values are marked by darker large dots. The bin size and $10^{th}$-$90^{th}$ percentile range are denoted by the horizontal and vertical bars, respectively. The gray line in each panel marks $\Gamma_\theta^F(\Gamma_S^F) \propto Re_b^{-1/2}$.**

## 5 Eddy diffusivities induced by turbulence and salt finger

### 5.1 Eddy diffusivities induced by turbulence

Since $\Gamma^T$ deviates from the conventionally used constant of 0.2 in the Osborn relation, $K_\rho^T$ (also $K_\theta^T$ and $K_S^T$) based on $\Gamma^T$ differs from $K_c$ based on 0.2 ($K_c = 0.2\varepsilon/N^2$) to different extents (Fig. 13). For the MIXET1, since $\Gamma^T$ is only slightly larger than 0.2 in general, the magnitudes of $K_\rho^T$ and $K_c$ differ slightly, with mean $K_c = 2.1 \times 10^{-6}$ m$^2$ s$^{-1}$ and mean $K_\rho^T = 4.6 \times 10^{-6}$ m$^2$ s$^{-1}$. Vertically, both $K_\rho^T$ and $K_c$ decrease in the upper 1200 m and increase at deeper depth. Obvious differences between $K_\rho^T$ and

$K_c$ occur at depth ranges shallower than 1200 m and deeper than 2000 m, where the mean ratio of $K_\rho^T$ to $K_c$ are 2.7 and 2.3, respectively. For the MIXET2, the magnitude difference between $K_\rho^T$ and $K_c$ is larger, with the mean values being $1.3 \times 10^{-6}$ m$^2$ s$^{-1}$ and $3.9 \times 10^{-6}$ m$^2$ s$^{-1}$, respectively. Compared with $K_c$ that stays nearly constant in the upper 1700 m, $K_\rho^T$ first decreases in the upper 700 m and then stays around $2 \times 10^{-6}$ m$^2$ s$^{-1}$ between 700 and 1700 m. For the BBTRE96, except for several depth bins, the difference between mean $K_\rho^T$ and mean $K_c$ in the upper 3700 m is small; and they share similar vertical increasing

rates and similar depth-averaged median values around $2.0 \times 10^{-5}$ m$^2$ s$^{-1}$, with $K_\rho^T$ is about 2.5 times of $K_c$. Although both increase at depths deeper than 3700 m, $K_\rho^T$ is nearly 4.7 times larger than $K_c$; and the mean values for $K_\rho^T$ and $K_c$ are $5.0 \times 10^{-4}$ and $1.1 \times 10^{-4}$ m$^2$ s$^{-1}$, respectively. For the BBTRE97, $K_\rho^T$ and $K_c$ share the same vertical decreasing trend and magnitude in the upper 1000 m, with mean values close to $2.6 \times 10^{-5}$ m$^2$ s$^{-1}$. Beneath 1000 m, although sharing similar increasing trend, $K_\rho^T$ becomes larger and larger than $K_c$ with depth. At depth between 1000 and 3700 m, $K_\rho^T/K_c \approx 2.7$ with median $K_\rho^T$ around





$8.3\times10^{-5}$ m$^2$ s$^{-1}$, while the corresponding values for depths deeper than 3700 m are $K_\rho^{T}/K_c\approx8.8$ and mean $K_\rho^{T}\approx1.2\times10^{-3}$ m$^2$ s$^{-1}$.

For the NATRE, $K_\rho^{T}$ is always larger than $K_c$ at all depth ranges; and the mean values of $K_\rho^{T}$ and $K_c$ are $4.4\times10^{-5}$ and $1.0\times10^{-5}$ m$^2$ s$^{-1}$, respectively. Vertically, $K_c$ generally fluctuating around its mean value for the whole water column. $K_\rho^{T}$ also shows no clear vertical trend in the upper 2700 m, but it increases significantly from $2.6\times10^{-5}$ m$^2$ s$^{-1}$ at 2700 m to $1.2\times10^{-4}$ m$^2$ s$^{-1}$ at 3900 m. As a result, $K_\rho^{T}$ is 13.7 times larger than $K_c$ at 3900 m. For the five projects, taking $\Gamma^{T}$ as a constant of 0.2

underestimates the actual eddy diffusivity induced by turbulence, and this underestimate may become more severe as $\Gamma^{T}$ increases with depth.

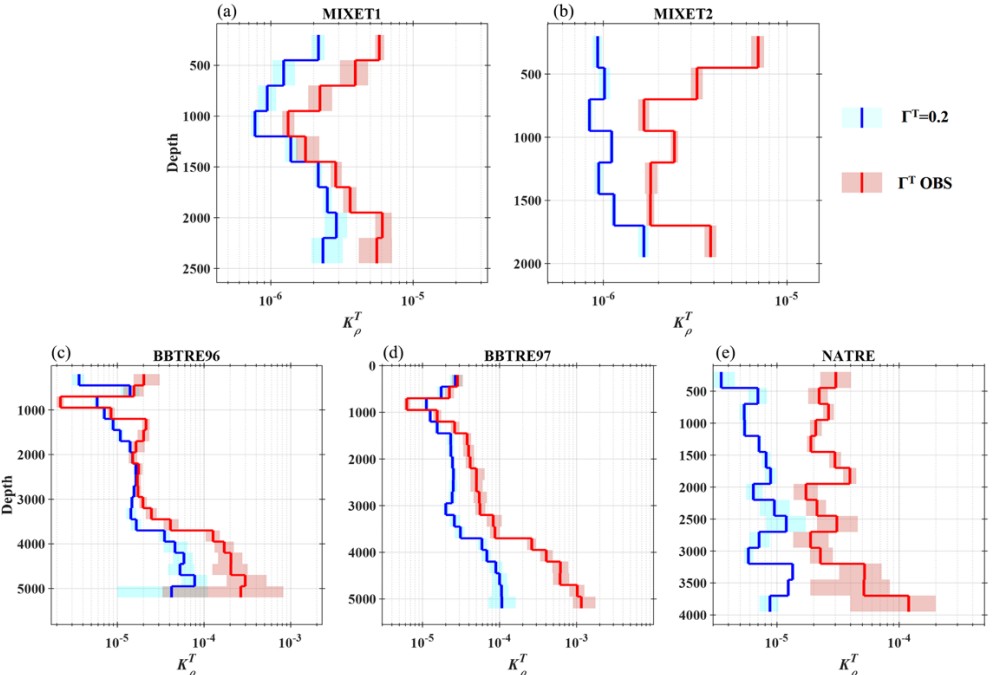

**Fig. 13. Vertical profiles of depth-bin mean $K_\rho^{T}(K_c)$ based on energetic turbulence and weak turbulence patches for the five projects. The blue curve is $K_c$ estimates by using $\Gamma^{T}=0.2$, and the red curve is $K_\rho^{T}$ based on the measured $\Gamma^{T}$. The colored shadings**

**correspond to 95% bootstrapped confidence intervals. To exclude the influence of extreme values, we only consider patches with $\Gamma^{T}$ within its upper and lower quartiles for each depth bin. The depth-bin size is 250 m.**

**5.2 Eddy diffusivities induced by salt finger**

For salt finger-induced eddy diffusivities, some studies estimated their values by taking a constant $r^{F}$ around 0.7 (0.75 in Schmitt et al., 2005; 0.6 in St. Laurent and Schmitt 1999). Here, $K_\theta^{F}$ derived from the observed $r^{F}$ is compared with the

$r^{F}=0.7$ estimate, $K_{\theta c}^{F}$ (Fig. 14). Depending on the deviation of the observed $r^{F}$ from 0.7, the five projects are distinct in terms of the difference between $K_\theta^{F}$ and $K_{\theta c}^{F}$. For the MIXET1, $K_\theta^{F}$ and $K_{\theta c}^{F}$ both vary little with depth. But the magnitude of $K_{\theta c}^{F}$ is significantly greater than that of $K_\theta^{F}$, with mean values being $2.2\times10^{-6}$ and $4.6\times10^{-7}$ m$^2$ s$^{-1}$, respectively. This is in line with



the fact that the mean value of the measured $r^F$ for the MIXET1 is only 0.37, about one half of 0.7. For MIXET2, with the median $r^F$ elevated to 0.63, $K_\theta^F{}_c$ is only slightly larger than $K_\theta^F$. And they both increase with depth form $O(10^{-6})$ m$^2$ s$^{-1}$ at 100 m to $O(10^{-5})$ m$^2$ s$^{-1}$ at 1850 m. The median values of $K_\theta^F{}_c$ and $K_\theta^F$ are $4.4\times10^{-6}$ m$^2$ s$^{-1}$ and $3.2\times10^{-6}$ m$^2$ s$^{-1}$, respectively. The difference between MIXET1 and MIXET2 indicates a strong seasonal variation of salt finger in the tropical Pacific. For both BBTRE96 and BBTRE97, $K_\theta^F{}_c$ is significantly larger than $K_\theta^F$ in the upper layer with magnitudes around $O(10^{-5})$ and $O(10^{-6})$ m$^2$ s$^{-1}$, respectively, and this difference turns small as they both increase to $2\times10^{-5}$ m$^2$ s$^{-1}$ with depth increasing to 2000 m. At deeper depths, although salt finger disappears at some depth ranges, $K_\theta^F{}_c$ varies little around $2.5\times10^{-5}$ m$^2$ s$^{-1}$. $K_\theta^F$ is generally larger than $K_\theta^F{}_c$ between 2400 m and 3400 m with $K_\theta^F/K_\theta^F{}_c$ varying between 3 and 10, and this ratio drops to less than 2 for depths deeper than 3400 m. For NATRE, both $K_\theta^F$ and $K_\theta^F{}_c$ present clear vertical increasing trends, and $K_\theta^F{}_c$ is dominantly greater than $K_\theta^F$. The difference between $K_\theta^F$ and $K_\theta^F{}_c$ is reduced with increasing depth, due to the fact that $K_\theta^F$ increases much faster in the vertical from about $2\times10^{-6}$ m$^2$ s$^{-1}$ at upper 500 m to $1.5\times10^{-5}$ m$^2$ s$^{-1}$ at 2400 m. For all the projects, $K_\theta^F$ is generally smaller than $K_\theta^F{}_c$ since $r^F$ is mostly smaller than 0.7; and this phenomenon is most obvious in the upper layer (upper 1000 m of the BBTRE96, BBTRE97, and NATRE). At deeper depths, $K_\theta^F>K_\theta^F{}_c$ can be observed in projects like the BBTRE96, BBTRE97. All these indicate $r^F$ is highly variable regionally and vertically. We also explore vertical variation of $K_S^F$, which is very similar to that of $K_\theta^F$ but with a larger magnitude (Fig. 15), as the result of $\Gamma_S^F$ being larger than and strongly proportional to $\Gamma_\theta^F$.

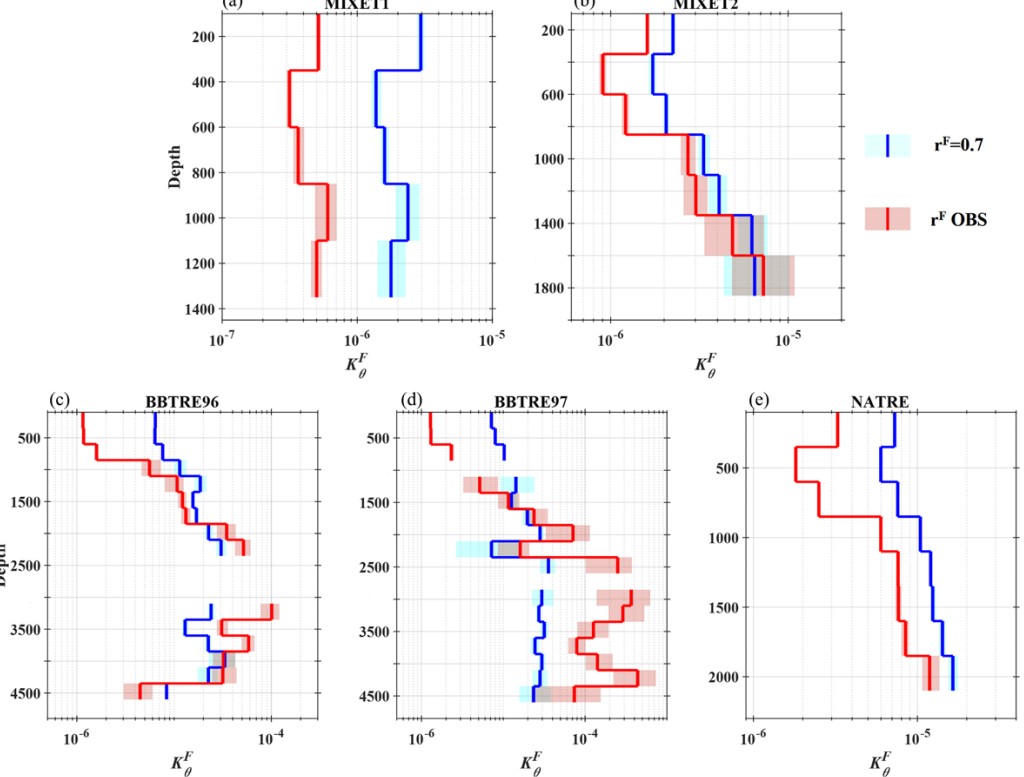



**Fig. 14. Vertical profiles of depth-bin mean $K_\theta^F(K_{\theta c}^F)$ based on salt finger patches for the five projects. The blue curves are $K_{\theta c}^F$ estimated with $r^F=0.7$, and the red ones are $K_\theta^F$ based on the measured $r^F$. The colored shades correspond to 95% bootstrapped confidence intervals. To exclude the influence of extreme values, we only consider patches with $\Gamma^T$ within its upper and lower quartiles for each depth bin. The depth-bin size is 250 m, and depth bins with patch number smaller than 10 are excluded.**

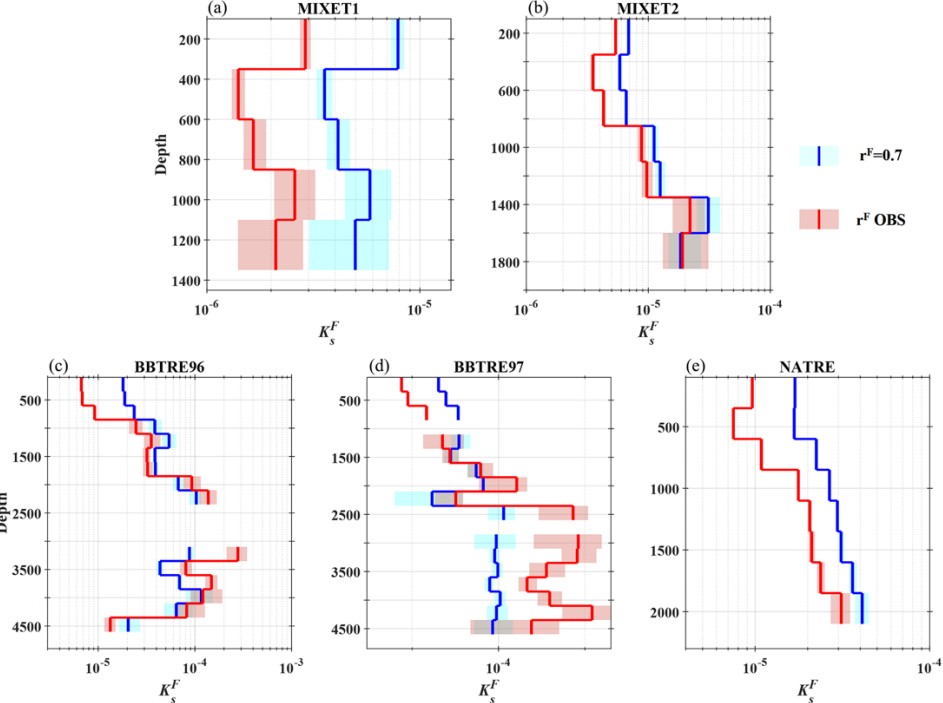


**Fig. 15. Same as Fig. 14, but for $K_S^F$.**

Next, we examine vertical variation of the ratio of $K_S^F$ to $K_\theta^F$ for the five projects (Fig. 16). For the MIXET1, $K_S^F/K_\theta^F$ generally decreases from 5.3 at upper 400 m to 4 at 1400 m, with an averaged value of 4.5. The averaged $K_S^F/K_\theta^F$ drops to 3.9 for the MIXET2, and it varies between 3.7 and 4.5 except the small values shallower than 400 m and beneath 1600 m.

The BBTRE projects share similar vertical structure of $K_S^F/K_\theta^F$: It has the maximum value of 6 in the upper 800 m, then sharply decreases to 2.5 at 1350 m and keeps at this value until reaching 4600 m. $K_S^F/K_\theta^F$ for the NATRE first increases from 3.0 to 4.7 in the upper 800 m, and then sharply decreases to 3 at 1100 m and remains unchanged. From the five projects, $K_S^F/K_\theta^F$ generally increases with depth at the upper 1000 m with an average value about 5; then, it sharply drops to around 3 and stays at this value at deeper depths. This ratio is reported to be 2.3 in the western tropical Atlantic (Schmitt et al., 2005),

slightly smaller than the result presented here. Van de Boog et al. (2021) presented a global map of $K_S^F$ and $K_\theta^F$ based on Argo data and an empirical method; and their results indicate $K_S^F/K_\theta^F$ vary between 1.3 and 7.8 for $R_\rho$ ranging from 1 to 4. These earlier works do not show the vertical variation of $K_S^F/K_\theta^F$ due to indirect methods used.





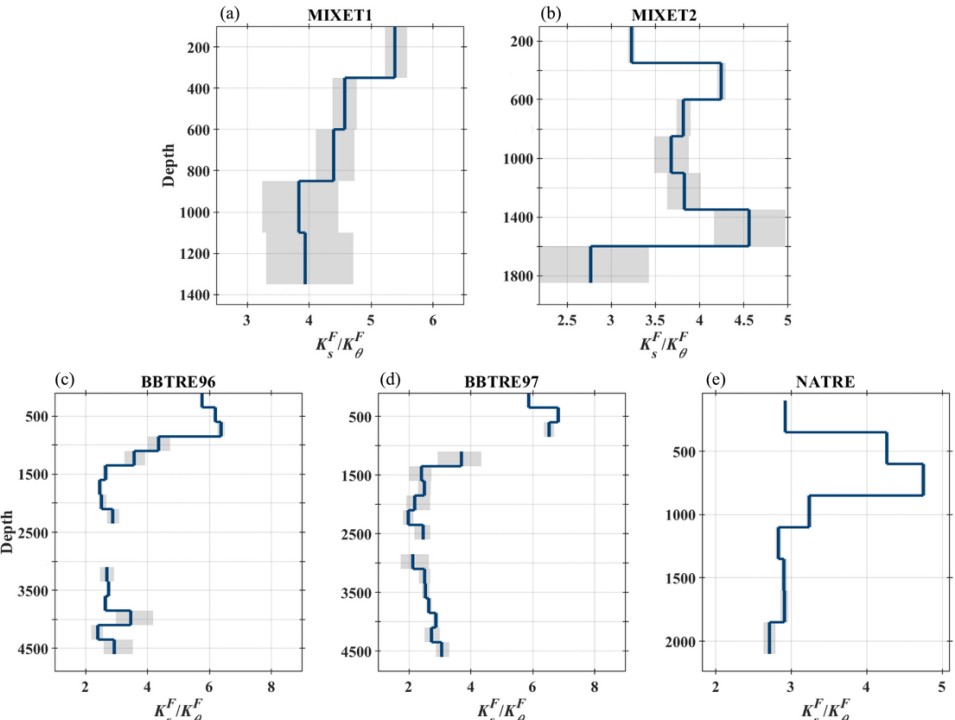

**Fig. 16. Vertical profiles of depth-bin mean $K_\theta^F/K_S^F$ based on salt finger patches for the five projects. The dark blue curves are the mean $K_\theta^F/K_S^F$, and the gray shadings are 95% bootstrapped confidence intervals. To exclude the influence of extreme values, we only consider patches with $\Gamma^T$ within its upper and lower quartiles for each depth bin. The depth-bin size is 250 m; and depth bins with patch number smaller than 10 are excluded.**

## 6 Summary

The Osborn relation is widely used to estimate vertical eddy diffusivity in practice, assuming a constant dissipation ratio of $\Gamma^T$=0.2 without identifying underlying mixing mechanisms. The dissipation ratios of heat, salinity and density are equal for turbulent mixing; however, they differ for salt finger-induced mixing. As a result, the eddy diffusivities derived from a constant dissipation ratio would inevitably depart from the actual values. In this study, we differentiated turbulent mixing and salt finger mixing, quantified their dissipation ratios and eddy diffusivities, and examined their relations based on the datasets from "Microstructure Database".

We evaluated the variation of $\Gamma^T$ and its relations with $Re_b$ and $R_{OT}$. The observed $\Gamma^T$ scatters over orders of magnitude, typically from $10^{-2}$ to 10. The significant difference between the five projects suggests $\Gamma^T$ is highly variable with space and time. Vertically, $\Gamma^T$ in the western equatorial Pacific presents a weak decreasing trend, while it increases obviously in the midlatitude in the Atlantic. Although a negative relation between $\Gamma^T$ and $Re_b$ was supported by most of the projects, further



investigation of the relations of $\Gamma^T$ with $Re_b$ and $R_{OT}$ suggested $\Gamma^T \propto R_{OT}^{-4/3} \cdot Re_b^{1/2}$. This indicates $\Gamma^T$ is modulated by more than one variable, and explains why different relations between $\Gamma^T$ and $Re_b$ have been reported (i.e., Mashayek et al., 2017; Ijichi and Hibiya, 2018).

We compared $K_\rho^T$ estimated using observed $\Gamma^T$ with $K_c$ estimated using $\Gamma^T=0.2$. $K_\rho^T$ is clearly larger than $K_c$. For the MIXET projects with vertically weak decreasing $\Gamma^T$, $K_\rho^T$ shares similar vertical structure of $K_c$, with magnitude elevated by about two or three times. For the rest projects whose $\Gamma^T$ increases significantly with depth, $K_\rho^T$ generally presents a much more obvious increasing trend than $K_c$, and $K_\rho^T$ can be larger than $K_c$ by an order of magnitude. This suggests the intensity of bottom-enhanced mixing may be underestimated when assuming $\Gamma^T=0.2$.

For salt finger, two "effective" dissipation ratios for heat ($\Gamma_\theta^F$) and salt ($\Gamma_S^F$) are derived, and two "artificial" Osborn relations are used to calculate corresponding eddy diffusivities. $\Gamma_\theta^F$ spans about two orders of magnitude. Both the magnitude and vertical structure of $\Gamma_\theta^F$ are distinct for the five projects. $\Gamma_S^F$ is strongly related to $\Gamma_\theta^F$, and they share similar vertical structures, $\Gamma_S^F \approx 5\Gamma_\theta^F$. Data from some projects indicate a negative relation between $\Gamma_\theta^F$ ($\Gamma_S^F$) and $Re_b$, while the others suggest no clear relation. Unlike the existing results indicating $r^F$ decreases then asymptotes to a constant value with increasing $R_\rho$, our results suggest $r^F$ decreases sharply with $R_\rho$ when it is smaller than 2.4 and grows to a larger value with $R_\rho$ when it exceeds 2.4.

We examined salt finger-induced $K_\theta^F$ and $K_S^F$. Although salt finger becomes rarer with depth, $K_\theta^F$ and $K_S^F$ increase clearly in the vertical, and $K_S^F$ is greater than $K_\theta^F$. In the upper 1000 m, $K_S^F$ is significantly greater than $K_\theta^F$ by about five times for most projects; but below 1000 m, $K_S^F/K_\theta^F$ generally stays around 3. $K_\theta^F$ and $K_S^F$ estimated using the observed $r^F$ are generally smaller than those using $r^F=0.7$ due to most observed $r^F$ being smaller than 0.7, but varying more sharply in the vertical.

Compared with eddy diffusivity induced by turbulence, $K_\theta^F$ is smaller than $K_\theta^T$ in the upper 1000 m, but they become more and more comparable with increasing depth. $K_S^F$ is close to or even larger than $K_S^T$ at all depths for all the projects. In general, although salt finger events are much rare than turbulence at depth (so they may be incapable of largely altering the background mixing intensity shaped by turbulence), they can play a crucial role in local, short-period mixing events, which is worth to be investigated and properly parameterized in numerical models.

*Data Availability*

The microstructure datasets used in this study are available at http://microstructure.ucsd.edu. And the ETOPO 2022 bathymetry data used in Fig. 1 is from https://www.ncei.noaa.gov/products/etopo-global-relief-model.

*Author contributions*



The study was conceived and designed by all co-authors. Data preparation, material collection, and analysis were performed by JL. JL prepared the manuscript with contributions from all co-authors.

*Competing interests*

The contact author has declared that none of the authors has any competing interests.

*Acknowledgments*

We thank the Climate Process Team for publicly sharing the "Microstructure Database". This work is supported by the National Natural Science Foundation of China (Grants 42076012, 42376012), the Postdoctoral Fellowship Program of CPSF
(Grant GZC20241610) and the China Postdoctoral Science Foundation (Grant 2024M753046).

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
