# Peer review of "Dissipation ratio and eddy diffusivity of turbulent and salt finger mixing derived from microstructure measurements"

_EGUsphere, 2024_

## Referee Comment (RC1)

**Review of "Dissipation ratio and eddy diffusivity of turbulent and salt finger mixing derived from microstructure measurements" by Li, Yang & Sun**

This manuscript takes microstructure measurements from a variety of different ocean basins with different propensities for salt finger double diffusive convection, and discusses the dissipation ratio (mixing efficiency $\Gamma$) and the turbulent diffusivities of heat, salt and buoyancy. However, I do not think that the salt-finger cases are treated properly, and so I recommend against this manuscript being published in *Ocean Science*.

This paper divides the observations into different sets based on the propensity to exhibit salt-fingering behaviour, as measured by the Turner angle. So far so good. But then the mixing efficiency, $\Gamma$, is estimated differently depending on which class of observations the measured data falls into. If the data comes from a doubly-stable regime, then the Oakey formula (which appears in between equations (2) and (3) of the manuscript)

$$\frac{\frac{1}{2}\chi_\theta N^2}{\epsilon\,\theta_z^2}$$

is used, whereas if the data is from a water column that has warm salty seawater overlying cooler fresher seawater, then a different formula is used, namely, from their Equations (4) and (5),

$$\Gamma_\theta^F = \left(\frac{R_\rho-1}{R_\rho}\right)\left(\frac{r^F}{1-r^F}\right) \qquad \text{and} \qquad \Gamma_S^F = \left(\frac{R_\rho-1}{1-r^F}\right).$$

We know from the careful study of St. Laurent and Schmitt (1999) that in the North Atlantic Central Water where $R_\rho$ is about 2 and so is susceptible to salt-fingering, the detection of salt fingers is very difficult. Their conclusion is that most of the time the observed microstructure is due to ordinary turbulent mixing which has the same turbulent diffusivity for all conserved scalar quantities. Hence, in such locations, it is not appropriate to assume that salt fingers account for all the observed microstructure, as the present manuscript assumes. This is the reason I recommend that the present manuscript should not be published in *Ocean Science*.

There is a way of using the microstructure observations while recognizing that they are the sum of contributions from both (1) isotropic turbulence and (2) salt fingering. This method appeared in section 3 of McDougall and Ruddick (1992), and it is quite different to what is used in the present manuscript.

Reference:
McDougall, T. J. and B. R. Ruddick, 1992: The use of ocean microstructure to quantify both turbulent mixing and salt-fingering. *Deep-Sea Research*, **39**, 1931-1952.

---

## Author Response (AR1)

**RESPONSE TO REVIEWERS**

**Manuscript Number: egusphere-2024-2749**

**Dissipation ratio and eddy diffusivity of turbulent and salt finger mixing derived from microstructure measurements**

Note: The reviewer's original comments are indicated in black, and our responses are indicated in blue. Our changes in the marked-up version of the revised manuscript are given in green.

**Responses to Reviewer #1**

This manuscript takes microstructure measurements from a variety of different ocean basins with different propensities for salt finger double diffusive convection, and discusses the dissipation ratio (mixing efficiency Γ) and the turbulent diffusivities of heat, salt and buoyancy. However, I do not think that the salt-finger cases are treated properly, and so I recommend against this manuscript being published in *Ocean Science*.

This paper divides the observations into different sets based on the propensity to exhibit salt-fingering behavior, as measured by the Turner angle. So far so good. But then the mixing efficiency, Γ, is estimated differently depending on which class of observations the measured data falls into. If the data comes from a doubly-stable regime, then the Oakey formula (which appears in between equations (2) and (3) of the manuscript)

$$\frac{\chi_\theta N^2}{2\varepsilon \theta_z^2}$$

is used, whereas if the data is from a water column that has warm salty seawater overlying cooler fresher seawater, then a different formula is used, namely, from their Equations (4) and (5),

$$\Gamma_\theta^{\mathrm{F}} = \left(\frac{R_\rho - 1}{R_\rho}\right)\left(\frac{r^{\mathrm{F}}}{1 - r^{\mathrm{F}}}\right) \text{ and } \Gamma_S^{\mathrm{F}} = \frac{R_\rho - 1}{1 - r^{\mathrm{F}}}.$$

We know from the careful study of St. Laurent and Schmitt (1999) that in the North Atlantic Central Water where $R_\rho$ is about 2 and so is susceptible to salt-fingering, the detection of salt fingers is very difficult. Their conclusion is that most of the time the observed microstructure is due to ordinary turbulent mixing which has the same turbulent diffusivity for all conserved scalar quantities. Hence, in such locations, it is not appropriate to assume that salt fingers account for all the observed microstructure, as the present manuscript assumes. This is the reason I recommend that the present manuscript should not be published in *Ocean Science*.

There is a way of using the microstructure observations while recognizing that they are the sum of contributions from both (1) isotropic turbulence and (2) salt fingering. This method appeared in section 3 of McDougall and Ruddick (1992), and it is quite different to what is used in the present manuscript.

Reference:
McDougall, T. J. and B. R. Ruddick, 1992: The use of ocean microstructure to quantify both turbulent mixing and salt-fingering. Deep-Sea Research, 39, 1931-1952.

**Responses**: We sincerely appreciate the reviewer's valuable comments on our manuscript. Here, we put our views on the reviewer's concerns, and hope they can satisfy the reviewer. The reviewer's concerns are addressed in detail as follows.

**Firstly, the reviewer has concerns about our estimates of the dissipation ratio, Γ, using different formulas for turbulent mixing and salt-finger mixing, respectively.** We are sorry for causing this confusion due to our inappropriate expression. Indeed, the dissipation ratio Γ has the same definition for turbulent mixing and salt-finger mixing as follows,

$$\Gamma = \frac{\chi_\theta N^2}{2\varepsilon\theta_z^2}. \tag{1}$$

Based on the production-dissipation balance for TKE (Osborn,1980)

$$(1 - R_f)K_\rho N^2 - R_f\varepsilon = 0, \tag{2}$$

and the production-dissipation balance for thermal variance (Osborn and Cox, 1972)

$$2K_\theta\theta_z^2 - \chi_\theta = 0, \tag{3}$$

we can get

$$\Gamma = \frac{\chi_\theta N^2}{2\varepsilon\theta_z^2} = \left(\frac{R_f}{1 - R_f}\right)\frac{K_\theta}{K_\rho}. \tag{4}$$

Combining the expressions of buoyancy $N^2$ and buoyancy flux $K_\rho N^2$ together (St. Laurent and Schmitt, 1999),

$$N^2 = g\alpha\theta_z(1 - 1/R_\rho), \tag{5}$$

$$K_\rho N^2 = g[\alpha(K_\theta\theta_z) - \beta(K_S S_z)] = g\alpha(K_\theta\theta_z)(1 - 1/r). \tag{6}$$

$K_\rho$ is derived as $K_\rho = K_\theta \frac{1-1/r}{1-1/R_\rho}$ and is substituted in Eq. (4), then, we finally obtain

$$\Gamma = \frac{\chi_T N^2}{2\varepsilon\theta_z^2} = \left(\frac{R_f}{1 - R_f}\right)\frac{K_\theta}{K_\rho} = \left(\frac{R_f}{1 - R_f}\right)\left(\frac{R_\rho - 1}{R_\rho}\right)\left(\frac{r}{r - 1}\right). \tag{7}$$

Here, we stress that Eq. (7) is applicable to both turbulent mixing and salt finger mixing. Some variables referred above are listed in Table R1.

For turbulent mixing only, $K_S = K_\theta = K_\rho$, and $r = R_\rho$, resulting in $\left(\frac{R_\rho-1}{R_\rho}\right)\left(\frac{r}{r-1}\right) = 1$. Then, Eq. (7) leads to

$$\Gamma = \frac{\chi_T N^2}{2\varepsilon\theta_z^2} = \frac{R_f}{1 - R_f}. \tag{8}$$

Since $R_f$ = B/P, $\Gamma$ can be written as $\Gamma = B/\varepsilon$ for steady and homogeneous turbulence, i.e., P=B+ $\varepsilon$. As a result, $\Gamma$ can be considered as the fraction of energy available to turbulent mixing, which helps mix different density waters essentially and gain the background potential energy. This is also the reason some literatures called $\Gamma$ mixing efficiency when turbulent mixing prevails, although it is somewhat misleading since $\Gamma$ can be greater than unity. Based on Eq. (2), we can get

$$K_\theta^T = K_S^T = K_\rho^T = \frac{R_f}{1 - R_f}\frac{\varepsilon}{N^2} = \Gamma^T \frac{\varepsilon}{N^2}, \tag{9}$$

where superscript "T" indicates turbulent mixing.

However, for salt finger mixing only, $\lim\limits_{P\to 0}\frac{R_f}{1 - R_f} = -1$. Then, Eq. (7) yields

$$\Gamma = \frac{\chi_T N^2}{2\varepsilon\theta_z^2} = -\frac{K_\theta}{K_\rho} = -\left(\frac{R_\rho - 1}{R_\rho}\right)\left(\frac{r}{r - 1}\right). \tag{10}$$

Based on Eq. (2), we can get

$$K_\rho^F = \frac{R_f}{1 - R_f}\frac{\varepsilon}{N^2} = -\frac{\varepsilon}{N^2}. \tag{11}$$

Using Eq. (10), $K_\theta$ can be obtained as follows,

$$K_\theta^F = \left(\frac{R_\rho - 1}{R_\rho}\right)\left(\frac{r}{1 - r}\right)\frac{\varepsilon}{N^2} = \Gamma_\theta^F \frac{\varepsilon}{N^2}. \tag{12}$$

Note that $\Gamma_\theta^F$ is the same as the expression of dissipation ratio in Eq. (10). Along with flux ratio, $r = \frac{K_\theta}{K_S}R_\rho$, $K_S$ can be written as follows,

$$K_S^F = \frac{R_\rho - 1}{1 - r}\frac{\varepsilon}{N^2} = \Gamma_S^F \frac{\varepsilon}{N^2}, \tag{13}$$

where superscript "F" indicates salt Finger mixing.

Now, the $\Gamma_\theta^F$ and $\Gamma_S^F$ in our manuscript are actually two **artificial** "mixing efficiencies", to make the estimations of $K_\theta^F$ and $K_S^F$ for salt finger mixing analogical to the Osborn relation for turbulent mixing. Identical "analogical" Osborn relation for salt finger mixing was developed in Schmitt et al. (2005). Here,

we call the terms $\left(\frac{R_\rho-1}{R_\rho}\right)\left(\frac{r}{1-r}\right)$ and $\frac{R_\rho-1}{1-r}$ before "$\varepsilon/N^2$" as $\Gamma_\theta{}^F$ and $\Gamma_S{}^F$, respectively. Investigating the statistic features of $\Gamma_\theta{}^F$ and $\Gamma_S{}^F$ can be practically useful when estimating $K_\theta{}^F$ and $K_S{}^F$ solely based on $\varepsilon$ and $N^2$.

**Table R1. Variables and their definitions.**

| Variable | Definition |
|---|---|
| P | Shear production term in the TKE equation |
| B | Buoyancy flux term in the TKE equation |
| $\chi_\theta$ | Dissipation rate of thermal variance |
| $N^2$ | Buoyancy frequency squared |
| $\varepsilon$ | Dissipation rate of TKE |
| $\theta_z$ | Vertical gradient of temperature |
| $S_z$ | Vertical gradient of salinity |
| $R_f$ | Flux Richardson number, $R_f$=B/P |
| $K_S, K_\theta, K_\rho$ | Eddy diffusivities of salt, heat, buoyancy |
| $\alpha, \beta$ | Expansion coefficient due to heat, contraction coefficient due to salinity |
| $R_\rho$ | Density ratio, $R_\rho=\alpha\theta_z/\beta S_z$ |
| $r$ | Density flux ratio, $r=\alpha K_\theta\theta_z/\beta K_S S_z=K_\theta/K_S\cdot R_\rho$. For salt finger, $r=\frac{R_\rho\Gamma}{R_\rho\Gamma+R_\rho-1}$ |

As suggested by the reviewer and Reviewer #2, we realized that the corresponding text, section 2.3 in the manuscript, was misleading; also, it was mostly a collection of some published literatures, not needed for the manuscript. Therefore, we reorganized section 2.3 in the revision as follows,

"Dissipation ratio $\Gamma$ is defined as
$$\Gamma = \frac{\chi_\theta N^2}{2\varepsilon\theta_z^2} \tag{1}$$
for turbulent mixing and salt-finger mixing (Oakey, 1985). Based on the production-dissipation balances for TKE and thermal variance (Osborn and Cox, 1972; Osborn,1980), and introducing $R_\rho$ and the density flux ratio $r=\alpha K_\theta\theta_z/\beta K_S S_z=K_\theta/K_S\cdot R_\rho$, we get
$$\Gamma = \frac{\chi_T N^2}{2\varepsilon\theta_z^2} = \left(\frac{R_f}{1-R_f}\right)\frac{K_\theta}{K_\rho} = \left(\frac{R_f}{1-R_f}\right)\left(\frac{R_\rho-1}{R_\rho}\right)\left(\frac{r}{r-1}\right), \tag{2}$$
which is applicable to both turbulent mixing and salt finger mixing (St. Laurent and Schmitt, 1999).

For turbulent mixing only, $K_S = K_\theta = K_\rho$. Then, Eq. (2) leads to
$$\Gamma^T = \frac{\chi_T N^2}{2\varepsilon\theta_z^2} = \frac{R_f}{1-R_f}, \tag{3}$$
and
$$K_\theta^T = K_S^T = K_\rho^T = \Gamma^T \frac{\varepsilon}{N^2}, \tag{4}$$
where superscript "T" indicates turbulent mixing.

However, for salt finger mixing only, with $\lim_{P\to 0}\frac{R_f}{1-R_f} = -1$ (St. Laurent and Schmitt, 1999), Eq. (2) yields
$$\Gamma^F = \frac{\chi_T N^2}{2\varepsilon\theta_z^2} = -\frac{K_\theta}{K_\rho} = -\left(\frac{R_\rho-1}{R_\rho}\right)\left(\frac{r}{r-1}\right), \tag{5}$$
which cannot be used directly to estimate the salt finger induced eddy diffusivities. And they are estimated separately by introducing $R_\rho$ and $r^F = R_\rho\Gamma^F/\left(R_\rho\Gamma^F + R_\rho - 1\right)$ (St. Laurent and Schmitt, 1999; Schmitt et al., 2005; Inoue et al., 2007),
$$K_\theta^F = \left(\frac{R_\rho-1}{R_\rho}\right)\left(\frac{r}{1-r}\right)\frac{\varepsilon}{N^2} = \Gamma_\theta^F \frac{\varepsilon}{N^2}, \quad K_S^F = \frac{R_\rho-1}{1-r}\frac{\varepsilon}{N^2} = \Gamma_S^F \frac{\varepsilon}{N^2}. \tag{6}$$

Note that all these equations are written into forms analogical to the Osborn relation for turbulent mixing. $\Gamma_\theta{}^F$ and $\Gamma_S{}^F$ are two artificial "mixing efficiencies", which are actually $\left(\frac{R_\rho-1}{R_\rho}\right)\left(\frac{r}{1-r}\right)$ and $\frac{R_\rho-1}{1-r}$ before "$\varepsilon/N^2$" for $K_\theta{}^F$ and $K_S{}^F$ estimation. $\Gamma_\theta{}^F$ is the same as $\Gamma^F$, while $\Gamma_S{}^F$ are further derived based on $R_\rho$ and $r^F$, $\Gamma_S{}^F=\Gamma^F\cdot R_\rho/r^F$. Investigating the statistic features of $\Gamma_\theta{}^F$ and $\Gamma_S{}^F$ can be practically useful when estimating $K_\theta{}^F$ and $K_S{}^F$ solely based on $\varepsilon$ and $N^2$." (**Lines 164-184 in the marked-up version of the revised manuscript.**)

We hope the reviewers find it improved, to be accurate and readable.

**Secondly, the reviewer thinks this study concludes that salt fingers account for all the observed microstructures in the North Atlantic Central Water and other locations where the data were collected.** We understand the reviewer's concern. However, we do not intend to examine the relative contributions of turbulent mixing and salt finger mixing in shaping the observed microstructures. Our focus is to investigate the necessity of differentiating mixing types and to refine their dissipation ratios on eddy diffusivity estimation. Therefore, based on their unique features, we only choose and analyze a part of the microstructure patches, which are overwhelmingly dominated by either turbulent mixing or salt finger mixing, and excluded the patches suspecting to be hybrids of different mixing types. In the manuscript, the statistical features of the "pure" turbulent patches and "pure" salt finger patches are examined separately in sections 4.1 (5.1) and 4.2 (5.2), respectively. We do not explore the relative contributions of these two mixing types. Taking the NATRE project conducted in the North Atlantic Central Water as an example, we chose patches with $|Tu|<45°$ or $|Tu|>90°$ as "pure" turbulent mixing, since double diffusion is prohibited in these situations. Meanwhile, we chose those with $60°<Tu<90°$, $Re_b<25$ and $|\chi_\theta|/|\varepsilon|\geq7$ as "pure" salt finger mixing, owing to the fact that all these criteria efficiently lower the possibility of the occurrence and intensity of turbulent mixing. Note that here we do not use $60°<Tu<90°$ as the sole criterion, since, as the reviewer suggested that although the North Atlantic Central Water where $R_\rho$ is about 2 and so is susceptible to salt-fingering, the detection of salt fingers is very difficult (St. Laurent and Schmitt, 1999). Among all the patches, only about 35% of them meet the above two criteria and are further analysed in terms of "pure" turbulent and salt-finger mixing, respectively (Fig. R1e, Table. R2). For the rest 65% of the microstructure patches of hybrid mixing types, although we excluded them from our analysis, we can reasonably infer that they are mostly dominated by turbulent mixing, as their $Re_b$ values exceed the typical range for salt finger. Besides, among the chosen patches, "pure" salt fingers are mostly confined in the upper 500 m (Fig. R2). Therefore, although it is not our focus, our results implicitly suggest turbulent mixing is undoubtedly the dominant contribution to the observed microstructure strength, in line with the conclusion drawn by St. Laurent and Schmitt (1999). The microstructures of the chosen five projects analysed in this study are all dominated by turbulent mixing (Fig. R1). Our concern here is the dissipation ratios and hence the eddy diffusivities separately induced by those chosen "pure" turbulent patches and "pure" salt finger patches. They are indeed a small fraction of the total observed microstructure patches, and their specific contributions to the total microstructure field are not covered by this manuscript.

[Figure]

**Fig. R1. Proportions of patches with different mixing types for each project: "pure" energetic turbulence, "pure" weak turbulence, "pure" salt finger, "pure" diffusive convection and hybrid (turbulence and salt**

**finger, or turbulence and diffusive convection). Patches with hybrid mixing types are excluded from the analysis.**

**Table. R2. Proportions of patches with different mixing types for each project: "pure" energetic turbulence, "pure" weak turbulence, "pure" salt finger, "pure" diffusive convection and hybrid (turbulence and salt finger, or turbulence and diffusive convection), and those for all projects. Patches with hybrid mixing types are excluded from the analysis.**

| | Proportion (%) | | | | |
|---|---|---|---|---|---|
| | energetic turbulence | weak turbulence | salt finger | diffusive convection | excluded hybrid |
| MIXET1 | 29.56 | 47.05 | 4.11 | 0.06 | 19.22 |
| MIXET2 | 2.48 | 51.48 | 11.32 | 0.16 | 34.56 |
| BBTRE96 | 6.55 | 33.31 | 5.91 | 0.53 | 53.70 |
| BBTRE97 | 8.67 | 38.19 | 4.56 | 0.08 | 48.5 |
| NATRE | 1.10 | 12.21 | 21.95 | 1.09 | 63.65 |
| All | 6.60 | 32.00 | 9.70 | 0.46 | 51.24 |

[Figure]

**Fig. R2. Vertical variation of normalized occurrence frequency of salt finger (SF), energetic turbulence (ET) and weak turbulence (WT) for NATRE. The occurrence frequency of each mixing type is normalized by its maximum, only reflecting its own vertical variation of prevalence, and cannot be compared with others.**

We are sorry that some expressions in the manuscript may have caused confusion. The major part is the first paragraph in section 3 (Lines 164-179), where we only introduced the small part of the chosen patches and didn't state the larger part of excluded ones clearly, would make the reader think we overemphasized the prevalence of salt finger, especially for the NATRE project. This paragraph has been revised to eliminate any misunderstanding statements, by adding two explanations in the revision: "For the BBTREs and NATRE, although a large proportion of the patches have $45°<Tu<90°$ and hence are salt finger favorable, most of them have elevated $Re_b$; thus, we infer these mixing events as hybrids of salt finger and turbulence but dominated by turbulence. These patches are excluded from analysis to highlight the difference between turbulent and salt finger mixing. Only a few patches are chosen as effective salt finger events. Therefore, it is turbulent mixing that dominates the observed microstructures, in line with the results based on the NATRE (St. Laurent and Schmitt, 1999)." We also add the text "Although dominated by turbulent mixing, the rest of the patches, more than 50%, are hybrids of different mixing types, and are excluded from the analysis." The phrase "The salt finger-induced eddy diffusivities become more comparable or even stronger than the turbulent diffusivities with depth" in the Abstract may also be misleading, which is rewritten now: "The salt finger-induced eddy diffusivities also increase with depth, with some being comparable to or even stronger than the mean turbulent ones." Besides, the proportions of the excluded patches with hybrid mixing types are listed in Table 2. **(Lines 20-22, Lines 164-184 and Lines 216-219 in the marked-up version of the revised manuscript.)**

**Moreover, as suggested by the editor, here we add the comparison of the methods used in our manuscript with those described in McDougall and Ruddick (1992).** We examined the "total" eddy diffusivities, $K_\theta$ and $K_s$, induced by superposed salt finger and turbulence by two different methods. The first is from McDougall and Ruddick (1992) (hereinafter MR92). MR92 does not need to differentiate salt finger and turbulent patches; it estimates the total eddy diffusivities by (i) evaluating the departure of observed $\Gamma$ ($\Gamma = \frac{\chi_\theta N^2}{2\varepsilon\theta_z^2}$) from a preset reasonable turbulent $\Gamma^T$ (e.g., $\Gamma^T$=0.265) and (ii) introducing a "salt flux enhancement factor", $M_0$, scaled by density ratio $R_\rho$ and buoyancy flux ratio $r$ (more details are given in McDougall and Ruddick (1992)). Here, $r$ is treated specifically depending on the mixing type, that is, $r^T=R_\rho$ for turbulence and $r^F=\frac{R_\rho\Gamma}{R_\rho\Gamma+R_\rho-1}$ for salt finger (St. Laurent and Schmitt, 1999).The second is from St. Laurent and Schmitt (1999) (hereinafter LS99), which differentiates turbulence and salt finger firstly, then estimates their eddy diffusivities separately, and finally obtains the total ones as $K_\theta = P^T \cdot K_\theta^T + P^F \cdot K_\theta^F$ and $K_s = P^T \cdot K_s^T + P^F \cdot K_s^F$, where $P^T$ and $P^F$ are the number proportions of turbulence and salt finger patches to their sum, respectively. The methods used in our original manuscript is similar to LS99, except that we focus on the differences between salt finger and turbulence, and hence we did not estimate the "total" eddy diffusivities contributed jointly by salt finger and turbulence.

Fig. R3 shows the "total" $K_\theta$ estimated by above two methods for the five projects. Compared with the BBTREs and NATRE, the results based on MR92 and LS99 present larger differences for MIXETs, which may be due to the fewer patches and more scattered $\Gamma^T$ and $\Gamma^F$ for MIXETs. Nonetheless, it is obvious that both estimates have similar magnitude and vertical trend for all the five projects. This suggests that both MR92 and LS99 can reasonably estimate the total eddy diffusivities when sat finger and turbulence coexist. Because the method used to estimate diffusivities in our manuscript is essentially the same as LS99, the consistency between LS99 and MR92 adds to our confidence in the conclusions reached in this study.

[Figure]

**Fig. R3. Vertical profiles of depth-bin averaged total $K_\theta$ based on turbulence and salt finger patches for the five projects. The blue curves are results based on MR92, and the red ones are based on LS99. The shades correspond to 95% bootstrapped confidence intervals. The depth-bin size is 250 m, and depth bins with number of patches smaller than 10 are excluded.**

Comparing the total $K_\theta$ with $K_\theta^T$ and $K_\theta^F$ (Figs 13, 14 in the manuscript, presented here as Figs. R4, R5), we can see $K_\theta$, especially for the LS99 result, is obviously closer to $K_\theta^T$ for all the five projects, confirming that

turbulence dominates the observed microstructures. This result addresses the reviewer's comment regarding "it is not appropriate to assume that salt fingers account for all the observed microstructure, as the present manuscript assumes" more thoroughly. We do note that $K_\theta$ in the upper 500 m for the BBTREs and NATRE are significantly lower than $K_\theta^T$, seemingly indicating a strong weakening of $K_\theta$ due to the prevalence of salt finger. However, the effect of salt finger is actually overestimated, since the dominant hybrid mixing patches at this depth range are all excluded, which should be dominated by turbulence, as indicated by the elevated $Re_b$. Therefore, the total $K_\theta$ should not be so weak in the upper 500 m for the BBTREs and NATRE, and should be much closer to $K_\theta^T$ if the hybrid mixing patches enter the analysis.

[Figure]

**Fig. R4. Vertical profiles of depth-bin mean $K_\rho^T(K_c)$ based on energetic turbulence and weak turbulence patches for the five projects. The blue curve is $K_c$ estimates by using $\Gamma^T=0.2$, and the red curve is $K_\rho^T$ based on the measured $\Gamma^T$. The shadings correspond to 95% bootstrapped confidence intervals. To exclude the influence of extreme values, we only consider patches with $\Gamma^T$ within its upper and lower quartiles for each depth bin. The depth-bin size is 250 m.**

[Figure]

**Fig. R5. Vertical profiles of depth-bin mean $K_\theta^F(K_{\theta}^{F}{}_c)$ based on salt finger patches for the five projects. The blue curves are $K_{\theta}^{F}{}_c$ estimated with $r^F=0.7$, and the red ones are $K_\theta^F$ based on the measured $r^F$. The shades correspond to 95% bootstrapped confidence intervals. To exclude the influence of extreme values, we only use patches with $\Gamma_\theta^F$ between its upper and lower quartiles for each depth bin. The depth-bin size is 250 m, and depth bins with number of patches smaller than 10 are excluded.**

Clearly, the total $K_s$ is very similar to the situation of $K_\theta$ (Fig. R6), and the only notable difference is that $K_s$ is not significantly weakened by salt finger in the upper 500 m for BBTREs and NATRE, owing to $K_s^F$ is clearly greater than $K_\theta^F$ and is much closer to $K_s^T$.

[Figure]

**Fig. R6. Same as Fig. R3, but for the total $K_S$.**

The main results presented above has been organized as a new subsection (Section 5.3) in the revised manuscript as following.

**"5.3 "Total" eddy diffusivities under superposed salt finger and turbulence**

We examine the "total" eddy diffusivities contributed by both salt finger and turbulence by combining the patches with weak turbulence, energetic turbulence and salt finger. Two different methods are used to estimate the "total" eddy diffusivities. The first is from McDougall and Ruddick (1992) (hereinafter MR92). MR92 does not need to differentiate salt finger and turbulent patches; it estimates the total eddy diffusivities by (i) evaluating the departure of observed $\Gamma$ (Eq. (1)) to a preset reasonable turbulent $\Gamma^T$ (e.g., $\Gamma^T$=0.265) and (ii) introducing a "salt flux enhancement factor", $M_0$, scaled by $R_\rho$ and $r$ (more details are given in McDougall and Ruddick (1992)). Here, $r$ is treated specifically depending on the mixing type, that is, $r^T=R_\rho$ for turbulence and $r^F=\frac{R_\rho\Gamma}{R_\rho\Gamma+R_\rho-1}$ for salt finger (St. Laurent and Schmitt, 1999). The second is from St. Laurent and Schmitt (1999) (hereinafter LS99), which differentiates turbulence and salt finger firstly, then estimates their eddy diffusivities separately, and finally obtains the total ones as $K_\theta = P^T \cdot K_\theta^T + P^F \cdot K_\theta^F$ and $K_s = P^T \cdot K_s^T + P^F \cdot K_s^F$, where $P^T$ and $P^F$ are the number proportions of turbulence and salt finger patches to their sum, respectively. Fig. 17 shows the "total" $K_\theta$ estimated by these two methods. Compared with the BBTREs and NATRE, the results based on MR92 and LS99 present larger differences for MIXETs, which may be due to the fewer patches and more scattered $\Gamma^T$ and $\Gamma^F$. Nonetheless, it is obvious that both estimates have similar magnitude and vertical trend for all the five projects. Comparing the total $K_\theta$ with $K_\theta^T$ and $K_\theta^F$ (Figs. 13, 14), we can see $K_\theta$, especially for the LS99 result, is obviously closer to $K_\theta^T$ for all the five projects, confirming that turbulence dominates the observed microstructures. Note that $K_\theta$ in the upper 500 m for the BBTREs and NATRE are significantly lower than $K_\theta^T$, seemingly indicating a strong weakening of $K_\theta$ due to the prevalence of salt finger. However, the effect of salt finger is actually overestimated, since the dominant hybrid mixing patches at this depth range are all excluded, which should be dominated by turbulence, as indicated by the elevated $Re_b$. The total $K_s$ is not shown since it is very similar to the situation of $K_\theta$, and the only notable difference is $K_s$ is not significantly weakened by salt finger in the upper 500 m for BBTREs and NATRE, owing to $K_s^F$ is clearly greater than $K_\theta^F$ and much closer to $K_s^T$ (Fig. 15).

[Figure]

Fig. 17. Vertical profiles of depth-bin averaged total $K_\theta$ based on turbulence and salt finger patches for the five projects. The blue curves are results based on MR92, and the red ones are based on LS99. The shades correspond to 95% bootstrapped confidence intervals. The depth-bin size is 250 m, and depth bins with number of patches smaller than 10 are excluded." **(Lines 523-547 and 654-655 in the marked-up version of the revised manuscript.)**

By reworking/reorganizing the aforementioned text, we hope the revised manuscript has addressed the reviewer's concerns clearly and meet the publication standard of *Ocean Science*.

Reference:

McDougall, T. J., and B. R. Ruddick, 1992: The use of ocean microstructure to quantify both turbulent mixing and salt-fingering. Deep Sea Research Part A. Oceanographic Research Papers, 39, 1931–1952, https://doi.org/10.1016/0198-0149(92)90006-F.

St. Laurent, L., and R. W. Schmitt, 1999: The contribution of salt fingers to vertical mixing in the North Atlantic Tracer Release Experiment. Journal of Physical Oceanography, 29, 1404–1424, https://doi.org/10.1175/1520-0485(1999)029<1404:tcosft>2.0.co;2.

**Responses to Reviewer #2**

Dissipation and related diffusion and mixing in the ocean are important for ocean dynamics (even global circulation) and for the transport of oceanic constituents. However, the causative processes are on very small scales (down to millimetres or less) as well as patchy and intermittent. Hence they are difficult to measure and quite impractical to model explicitly. As a result, empirical relationships and parameterizations are much used and evidence to improve these is valuable. This manuscript especially concerns the much used factor $\Gamma^T$ = 0.2 in the Osborn relation for turbulent density diffusivity = $\Gamma^T$ (dissipation)/(buoyancy frequency$^2$). Much evidence is cited that $\Gamma^T$ varies a lot in space and time. Moreover, turbulence is not the only agent of mixing; for salt finger mixing the equivalent $\Gamma^F$ may be negative. The manuscript provides separate evidence and discussion of the effective diffusivities for temperature and salinity associated with salt fingering.

**Response**: We sincerely thank the reviewer for the valuable comment, which helps a lot to improve the quality of our manuscript. We hope the revised manuscript meet the reviewer's requirements, and we expect this work make a positive contribution to better interpret microstructure observations and favour the improvements of mixing parameterizations.

Data for the study are from the western equatorial Pacific the subtropical NE Atlantic and tropical SW Atlantic. This is a varied set but I am left uncertain as to how comprehensive or representative it may be of all the possible data that might have been used. It is certainly sufficient to make the case that there can be improvement by moving to $\Gamma^T$ other than 0.2 with some suggestion of how to derive improved values.

**Response**: We understand the reviewer's concern. The data used in this study is from the "Microstructure Database" (MacKinnon et al., 2017), which is publicly shared and constantly updated, and consists of most known microstructure observation projects. We chose all projects that meet the analysis requirement for this study. Before this study, we examined the dissipation ratio based on microstructure data obtained from the South China Sea (Li et al., 2023), which suggests similar relations between $\Gamma^T$ and $R_{OT}$ and $Re_b$, indicating the $\Gamma^T$ formula in this study could be representative. This discussion has been mentioned in the revised manuscript. **(Lines 104-106 in the marked-up version of the revised manuscript.)**

Reference:

MacKinnon, J. A., and Coauthors, 2017: Climate Process Team on Internal Wave–Driven Ocean Mixing. Bulletin of the American Meteorological Society, 98, 2429–2454, https://doi.org/10.1175/BAMS-D-16-0030.1.

Li, J., Yang, Q., Sun, H., Zhang, S., Xie, L., Wang, Q., Zhao, W., and Tian, J., 2023: On the Variation of Dissipation Flux Coefficient in the Upper South China Sea, J. Phys. Oceanogr., 53, 551–571, https://doi.org/10.1175/JPO-D-22-0127.1.

I found section 2.3 lacking in logical development and am unsure as to its value to the rest of the manuscript (it is rarely referred to). Might it be replaced by a few literature references?

**Response**: we thank the reviewer for this valuable suggestion. In the section 2.3, we intended to give a detail introduction of the derivation of $\Gamma$ and eddy diffusivities for both turbulent mixing and salt finger mixing. However, as suggested by the reviewer, we realized that this part is lacking in logical development and is rarely referred to in the whole manuscript as the reviewer pointed out.

Therefore, we simplified and reworked it as:

"Dissipation ratio $\Gamma$ is defined as

$$\Gamma = \frac{\chi_\theta N^2}{2\varepsilon\theta_z^2} \tag{1}$$

for turbulent mixing and salt-finger mixing (Oakey, 1985). Based on the production-dissipation balances for TKE and thermal variance (Osborn and Cox, 1972; Osborn,1980), and introducing $R_\rho$ and the density flux ratio $r = \alpha K_\theta \theta_z / \beta K_S S_z = K_\theta / K_S \cdot R_\rho$, we get

$$\Gamma = \frac{\chi_T N^2}{2\varepsilon \theta_z^2} = \left(\frac{R_f}{1-R_f}\right) \frac{K_\theta}{K_\rho} = \left(\frac{R_f}{1-R_f}\right)\left(\frac{R_\rho - 1}{R_\rho}\right)\left(\frac{r}{r-1}\right), \tag{2}$$

which is applicable to both turbulent mixing and salt finger mixing (St. Laurent and Schmitt, 1999). For turbulent mixing only, $K_S = K_\theta = K_\rho$. Then, Eq. (2) leads to

$$\Gamma^T = \frac{\chi_T N^2}{2\varepsilon \theta_z^2} = \frac{R_f}{1-R_f}, \tag{3}$$

and

$$K_\theta^T = K_S^T = K_\rho^T = \Gamma^T \frac{\varepsilon}{N^2}, \tag{4}$$

where superscript "T" indicates turbulent mixing.

However, for salt finger mixing only, with $\lim\limits_{P\to 0} \frac{R_f}{1-R_f} = -1$ (St. Laurent and Schmitt, 1999), Eq. (2) yields

$$\Gamma^F = \frac{\chi_T N^2}{2\varepsilon \theta_z^2} = -\frac{K_\theta}{K_\rho} = -\left(\frac{R_\rho - 1}{R_\rho}\right)\left(\frac{r}{r-1}\right), \tag{5}$$

which cannot be used directly to estimate the salt finger induced eddy diffusivities. And they are estimated separately by introducing $R_\rho$ and $r^F = R_\rho \Gamma^F / (R_\rho \Gamma^F + R_\rho - 1)$ (St. Laurent and Schmitt, 1999; Schmitt et al., 2005; Inoue et al., 2007),

$$K_\theta^F = \left(\frac{R_\rho - 1}{R_\rho}\right)\left(\frac{r}{1-r}\right)\frac{\varepsilon}{N^2} = \Gamma_\theta^F \frac{\varepsilon}{N^2}, \quad K_S^F = \frac{R_\rho - 1}{1-r}\frac{\varepsilon}{N^2} = \Gamma_S^F \frac{\varepsilon}{N^2}. \tag{6}$$

Note that all these equations are written into forms analogical to the Osborn relation for turbulent mixing. $\Gamma_\theta^F$ and $\Gamma_S^F$ are two artificial "mixing efficiencies", which are actually $\left(\frac{R_\rho - 1}{R_\rho}\right)\left(\frac{r}{1-r}\right)$ and $\frac{R_\rho - 1}{1-r}$ before "$\varepsilon/N^2$" for $K_\theta^F$ and $K_S^F$ estimation. $\Gamma_\theta^F$ is the same as $\Gamma^F$, while $\Gamma_S^F$ are further derived based on $R_\rho$ and $r^F$, $\Gamma_S^F = \Gamma^F \cdot R_\rho / r^F$. Investigating the statistic features of $\Gamma_\theta^F$ and $\Gamma_S^F$ can be practically useful when estimating $K_\theta^F$ and $K_S^F$ solely based on $\varepsilon$ and $N^2$." **(Lines 164-184 in the marked-up version of the revised manuscript.)**

We hope the reviewer find this revision more readable.

References:
Inoue, R., H. Yamazaki, F. Wolk, T. Kono, and J. Yoshida, 2007: An Estimation of Buoyancy Flux for a Mixture of Turbulence and Double Diffusion. Journal of Physical Oceanography, 37, 611–624, https://doi.org/10.1175/JPO2996.1.

Oakey, N. S., 1985: Statistics of Mixing Parameters in the Upper Ocean During JASIN Phase 2. Journal of Physical Oceanography, 15, 1662–1675, https://doi.org/10.1175/1520-0485(1985)015<1662:SOMPIT>2.0.CO;2.

Osborn, T. R., 1980: Estimates of the Local Rate of Vertical Diffusion from Dissipation Measurements. Journal of Physical Oceanography, 10, 83–89, https://doi.org/10.1175/1520-0485(1980)010<0083:EOTLRO>2.0.CO;2.

Osborn, T. R., and C. S. Cox, 1972: Oceanic fine structure. Geophysical Fluid Dynamics, 3, 321–345, https://doi.org/10.1080/03091927208236085.

Schmitt, R. W., J. R. Ledwell, E. T. Montgomery, K. L. Polzin, and J. M. Toole, 2005: Enhanced Diapycnal Mixing by Salt Fingers in the Thermocline of the Tropical Atlantic. Science, 308, 685–688, https://doi.org/10.1126/science.1108678.

St. Laurent, L., and R. W. Schmitt, 1999: The contribution of salt fingers to vertical mixing in the North Atlantic Tracer Release Experiment. Journal of Physical Oceanography, 29, 1404–1424, https://doi.org/10.1175/1520-0485(1999)029<1404:tcosft>2.0.co;2.

The English is generally understandable but there is some curious usage that should be corrected by the publisher's copy-editing of a final manuscript. The authors unfortunately follow the current "fashion" of misleadingly using "increasing trend" when they mean "positive trend" or simply "increase". "Misleading" because the expression implies a change of trend. [Many but probably not all examples are included in the following "Detailed comments".]

**Response**: We thank the reviewer for point out this misleading usage. We tried our best to polish the language in the whole revised manuscript. All changes are marked in the revised manuscript one by one, especially for the misleadingly usage of "trend". We have thoroughly checked and revised all "trend" used in the manuscript. **(All revisions of the usage of "trend" are highlighted in the marked-up version of the revised manuscript.)**

Detailed comments

Line 104. "we chose five projects that . ." Did other projects provide $\chi_\theta$ and you chose not to use them, or did you use all the projects providing $\chi_\theta$? If the latter, better "we chose all five projects that . ." to show that you did the best possible.

**Response**: We apologize for this unclear expression. The data used in this study should meet two criteria. First, $\chi_\theta$ is available to estimate $\Gamma$. Secondly, variables should be sampled and provided in the form of vertical profile, since vertical gradients of some variables (like $\theta$ and $S$) are needed. We have chosen all five projects meeting these two criteria.

This phrase is revised now as "Since the calculation of dissipation ratio requires the dissipation rate of thermal variance ($\chi_\theta$), and the vertical gradients of temperature $\theta$ and salinity $S$ are needed, we chose all five projects that provide $\chi_\theta$ and are in the form of vertical profiles." **(Lines 104-106 in the marked-up version of the revised manuscript.)**

Table 1. According to figure 1 NATRE is in the North Atlantic. ("S" –> "N").

**Response**: We are sorry for this error. It has been corrected. **(Line 112 in the marked-up version of the revised manuscript.)**

"Profile Number" –> "Number of Profiles".

**Response**: This phrase has been corrected. **(Line 112 in the marked-up version of the revised manuscript.)**

Equations (3). The sequence from left to right is not logical if $\Gamma$ is already defined as in line 149. Moreover from (1) and (2) the second and last terms of (3) are directly equal irrespective of the first and third terms. Please clarify what is definition, what is derivation, and where the form of the third term comes from (there seems to be an analogy with the right-hand side of (4) for which a reference is cited).

**Response**: We apologize for this illogical presentation. As aforementioned, due to the confusing expression and the weak connection to the following text, the section 2.3 has been reorganized and reworked. **(Lines 164-184 in the marked-up version of the revised manuscript.)**

Lines 212-213. Better ". . divide the number of energetic turbulence patches in each depth bin by the total number of energetic turbulence patches in the whole project; . ." and Line 214 ". . by the total number of patches within the same depth bin . ."?

**Response**: Thanks for this wording correction. The corresponding text has been revised. **(Lines 244-246 in the marked-up version of the revised manuscript.)**

Section 4.1 is very long and I think would benefit from some sub-headings.

**Response**: We thank the reviewer for this nice suggestion. Section 4.1 now consists of two subsections, namely "4.1.1 Vertical variation" and "4.1.2 Relation between $\Gamma^T$, $Re_b$ and $R_{OT}$". **(Lines 267 and 312 in the marked-up version of the revised manuscript.)**

Lines 243-244. Why are there two NATRE median values of $\Gamma^T$ for each of energetic turbulence and weak turbulence?

**Response**: We apologize for this wording issue. It has been corrected as "the median $\Gamma^T$ values are 0.33 and 0.50 for energetic turbulence and weak turbulence, respectively." **(Lines 276-277 in the marked-up version of the revised manuscript.)**

Line 257. Why "alternately"? "slightly increasing trend" –> "slight increase with depth"? (unless you mean the trend/rate increases).

**Response**: We are sorry for this wrong usage of "trend". It has been revised. **(Line 290 in the marked-up version of the revised manuscript.)**

Lines 268, 270, 451, 498, 510. "vertical increase" (or decrease") is unclear until upwards or downwards is specified. Also (lines 268, 270, 451) I think you mean "increase" not "increasing trend" (c.f. line 257; does the trend/rate increase?).

**Response**: We apologize for these improper wordings. "Vertical increase" is actually "increase downwards". These errors have been corrected. **(All necessary revisions of the usage of "vertical" are highlighted in the marked-up version of the revised manuscript.)**

Line 272. I think you mean ". . disagree about whether $\Gamma^T$ is larger for energetic turbulence or weak turbulence."

**Response**: We are sorry for this confusing expression. It has been revised. **(Line 306 in the marked-up version of the revised manuscript.)**

Lines 284, 369, 390, 451, 500. "increasing trend" –> "increase" (indeed, in line 390 referring to figure 11, the trend is positive but actually decreases for $R_\rho > 3$).

**Response**: We apologize for these errors. they have been corrected together. **(All necessary revisions of the usage of "trend" are highlighted in the marked-up version of the revised manuscript.)**

Lines 285, 290, 397. "decreasing trend" –> "decrease".

**Response**: It has been corrected. **(All necessary revisions of the usage of "trend" are highlighted in the marked-up version of the revised manuscript.)**

Line 286. "BBTRES" -> "BBTREs".

**Response**: It has been corrected. **(Line 322 in the marked-up version of the revised manuscript.)**

Figure 8. In the figure legend, the red line should be ascribed to $Re_b > 160$. The grey dashed line is rather indistinct.

**Response**: We are sorry for these mistakes. Fig. 8 has been revised as suggested, presented here as Fig. R7 for your information.

[Figure]

**Fig. R7. Relation between overturn-based $\Gamma^T$ and $R_{OT}$, overturns from the five projects are considered. The shading describes the distribution of probability density, with yellow indicating minimum probability density and blue representing maximum one. The overturns are correspondingly divided into two clusters: the gray dots have $Re_b<160$, and the pink ones, $Re_b>160$. The black and red lines represent $\Gamma^T \propto R_{OT}^{-4/3}$, crossing the centers of the two clusters. The white dashed line is the general relation between $\Gamma^T$ and $R_{OT}$ of the whole data collection.**

**(Lines 361 and 365 in the marked-up version of the revised manuscript.)**

Figure 9. Please explain (in the caption or against the colour bar) that the colour bar refers to median $\Gamma^T$.

**Response**: We apologize for this information gap. We added explains about the colorbar in the caption. **(Lines 381-382 in the marked-up version of the revised manuscript.)**

Line 372. The "ref"erence needs to be included.

**Response**: We are sorry for this mistake. This reference has been referred to correctly. **(Line 411 in the marked-up version of the revised manuscript.)**

Line 376. "Note that . . ." I think this sentence should refer to a labelled formula in section 2.3.

**Response**: We thank the reviewer for this suggestion. This sentence now refers to equation (6) in the revised manuscript. **(Line 415 in the marked-up version of the revised manuscript.)**

Line 403. "decreasing rate" –> "rate of decrease"

**Response**: Sorry for this wording issue. It has been corrected. **(Line 443 in the marked-up version of the revised manuscript.)**

Lines 419-420. "increasing rates" –> "increases"?

**Response**: It has been revised. **(Lines 459-460 in the marked-up version of the revised manuscript.)**

Line 422. "vertical decreasing trend and magnitude" –> "decrease downwards"?

**Response**: This phrase has been corrected as suggested. **(Lines 462-463 in the marked-up version of the revised manuscript.)**

Line 423. "increasing trend" –> "increases".

**Response**: It has been corrected. **(Line 464 in the marked-up version of the revised manuscript.)**

Line 492. "Vertically, $\Gamma^T$ in the western equatorial Pacific presents a weak decreasing trend" $\rightarrow$ "$\Gamma^T$ in the western equatorial Pacific presents a weak decrease downwards".

**Response**: We thank the reviewer very much for helping us polishing language. This sentence has been reworked. **(Line 557 in the marked-up version of the revised manuscript.)**

---

## Author Response (AR2)

**RESPONSES TO REVIEWERS AND EDITOR**

**Manuscript Number: egusphere-2024-2749**

**Dissipation ratio and eddy diffusivity of turbulent and salt finger mixing derived from microstructure measurements**

Note: The reviewer/editor's original comments are indicated in black, and our responses are indicated in blue. Our changes in the marked-up version of the revised manuscript are given in green.

**Responses to Reviewer #2**

There is some unusual use of English but I think the intended meaning is clear enough to be left to copy-editing.

**Response**: We thank the reviewer for pointing out this. To ensure accuracy and clarity, we carefully reviewed and refined the entire manuscript again, with additional polishing by native English speakers. We hope the reviewer find the revision much improved. **(All revisions are highlighted in the marked-up version of the revised manuscript.)**

Line 122. Please check the formulae in ($Tu$=atan$^{-1}$($\alpha\theta_z$-$\beta S_z$, -$\alpha\theta_z$+$\beta S_z$)) for the Turner angle. The second is exactly the negative of the first which would imply 45° or -45° only.

**Response**: The reviewer is right, and we are sorry for this mistake. The formulae used to calculate the Turner angle is $Tu$=atan$^{-1}$(-$\alpha\theta_z$-$\beta S_z$, -$\alpha\theta_z$+$\beta S_z$). It has been corrected in the revised manuscript. **(Line 122 in the marked-up version of the revised manuscript.)**

Section 2.3. I think this is now logical but not easily followed by myself as a reader less familiar with the quantities involved. The steps in (2) need reference either to the papers cited or to the authors' response to the first round of reviews. Lines 178-180 "introduce" r$^F$ which does not then appear in (6). [Was the superscript omitted to avoid "clutter"? Is a separate symbol r$^F$ necessary? I see that r$^F$ is used in section 4.2]

**Response**: We thank the reviewer for this helpful suggestion. St. Laurent and Schmitt (1999) provided the detailed derivation of the Eq. (2), $\Gamma = \frac{\chi_T N^2}{2\varepsilon\theta_z^2} = \left(\frac{R_f}{1-R_f}\right)\frac{K_\theta}{K_\rho} = \left(\frac{R_f}{1-R_f}\right)\left(\frac{R_\rho-1}{R_\rho}\right)\left(\frac{r}{r-1}\right)$, and we added it as the reference of the steps in Eq. (2). We introduce $r^F$ because only density flux ratio of salt finger can be derived using $r^F = R_\rho\Gamma^F/(R_\rho\Gamma^F + R_\rho - 1)$, and $r$ in the Eq. (6), $K_\theta^F = \left(\frac{R_\rho-1}{R_\rho}\right)\left(\frac{r}{1-r}\right)\frac{\varepsilon}{N^2} = \Gamma_\theta^F\frac{\varepsilon}{N^2}, K_S^F = \frac{R_\rho-1}{1-r}\frac{\varepsilon}{N^2} = \Gamma_S^F\frac{\varepsilon}{N^2}$, should be $r^F$. We corrected this mistake in the revised manuscript. **(Lines 153, 154, 161, 164 and 166 in the marked-up version of the revised manuscript.)**

Figure 16. The horizontal axes are labelled as (salinity diffusivity) / (temperature diffusivity), consistent with text lines 507-517, but the caption (lines 519, 520) refers to the inverse ratio.

**Responses**: We thank the reviewer for pointing out this mistake. It has been corrected. **(Lines 496, 497 in the marked-up version of the revised manuscript.)**

**Responses to the handling editor**

Line 107 Change 'ect' to " and other information".

**Response**: We thank the handling editor for this suggestion. It has been revised in the revision. **(Line 107 in the marked-up version of the revised manuscript.)**